# SPARKE: Scalable Prompt-Aware Diversity and Novelty Guidance in Diffusion Models via RKE Score

**Mohammad Jalali**[1][*]  **Haoyu Lei**[1][*]  **Amin Gohari**[2]  **Farzan Farnia**[1]

[1]Department of Computer Science and Engineering, The Chinese University of Hong Kong
[2]Department of Information Engineering, The Chinese University of Hong Kong
{mjalali24, hylei22, farnia}@cse.cuhk.edu.hk, agohari@ie.cuhk.edu.hk

## Abstract

Diffusion models have demonstrated remarkable success in high-fidelity image synthesis and prompt-guided generative modeling. However, ensuring adequate diversity in generated samples of prompt-guided diffusion models remains a challenge, particularly when the prompts span a broad semantic spectrum and the diversity of generated data needs to be evaluated in a prompt-aware fashion across semantically similar prompts. Recent methods have introduced guidance via diversity measures to encourage more varied generations. In this work, we extend the diversity measure-based approaches by proposing the _**S**calable **P**rompt-**A**ware **R**ény **K**ernel **E**ntropy Diversity Guidance_ (_SPARKE_) method for prompt-aware diversity guidance. SPARKE utilizes conditional entropy for diversity guidance, which dynamically conditions diversity measurement on similar prompts and enables prompt-aware diversity control. While the entropy-based guidance approach enhances prompt-aware diversity, its reliance on the matrix-based entropy scores poses computational challenges in large-scale generation settings. To address this, we focus on the special case of _Conditional latent RKE Score Guidance_, reducing entropy computation and gradient-based optimization complexity from the $\mathcal{O}(n^3)$ of general entropy measures to $\mathcal{O}(n)$. The reduced computational complexity allows for diversity-guided sampling over potentially thousands of generation rounds on different prompts. We numerically test the SPARKE method on several text-to-image diffusion models, demonstrating that the proposed method improves the prompt-aware diversity of the generated data without incurring significant computational costs. We release our code on the project page: https://mjalali.github.io/SPARKE.

## 1  Introduction

Diffusion models [1, 2, 3] have rapidly become a prominent class of generative models, achieving state-of-the-art results in several generative modeling tasks [2, 4, 5]. Notably, their ability to produce intricate and realistic image and video data has significantly advanced various content creation pipelines [6, 7, 8, 9]. Despite these successes, ensuring sufficient diversity in generated outputs remains an area of active research, particularly in prompt-guided diffusion models, where the diversity of generated samples needs to be evaluated and optimized while considering the variability of the input prompts.

Diversity in prompt-guided sample generation also plays a critical role in ensuring fairness and lack of mode collapse in response to a certain prompt category, implying that the outputs do not collapse in response to a group of prompts with similar components. Therefore, an adequately diverse

---

[*]Contributed Equally

39th Conference on Neural Information Processing Systems (NeurIPS 2025).

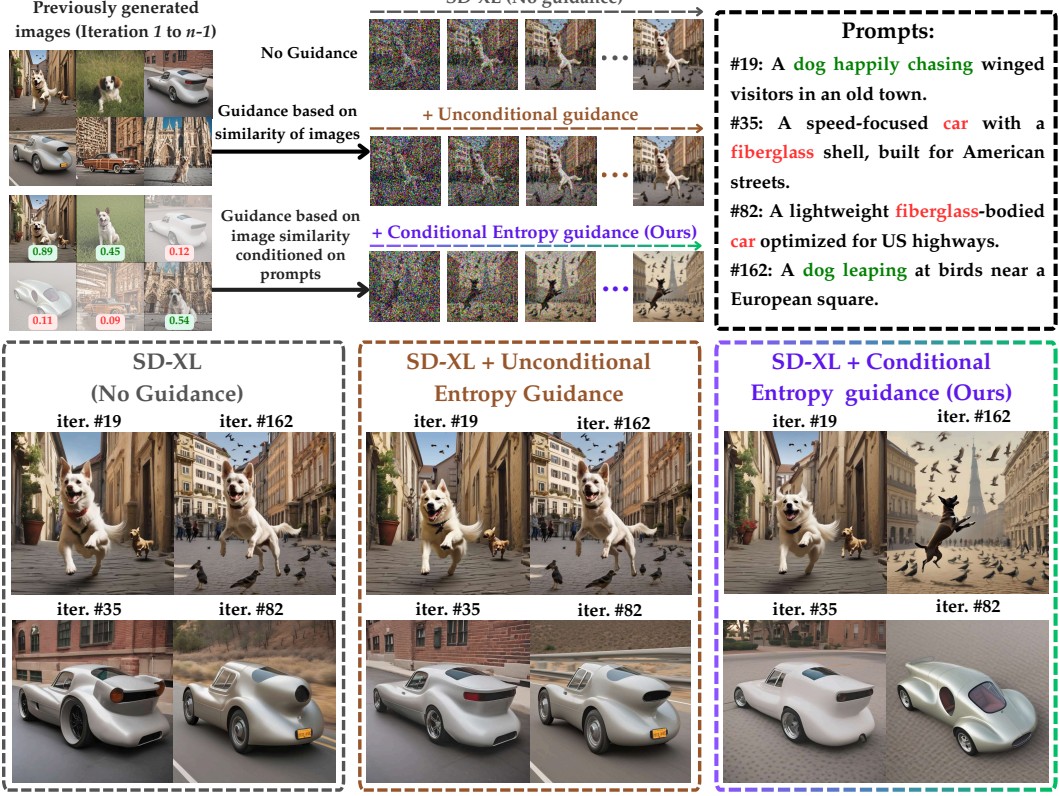

Figure 1: Overview of the proposed SPARKE method in generating images at different iterations in comparison to the vanilla Stable Diffusion-XL [5] model. We also compare the conditional-RKE guidance in SPARKE with the baseline Vendi Score guidance [10] (unconditional, in latent space).

prompt-guided generation model is supposed to provide varied and balanced outputs conditioned on each prompt category. For example, in geographically diverse image synthesis, the generation model is supposed to represent a broad spectrum of regional styles, preventing overrepresentation of dominant patterns while ensuring fair coverage across different locations [10]. Similarly, in data augmentation for visual recognition, generating a semantically varied set of samples enhances the model's generalizability and reduces biases in prompt-guided sample creation [11, 12].

A recently explored strategy for diversity-aware generation in diffusion models is to utilize a diversity score for guiding the sample generation. One such method is the contextualized Vendi score guidance (c-VSG) [10]. The c-VSG approach builds upon the Vendi diversity measure [13], which is a kernel matrix-based entropy score evaluating diversity in generative models. The numerical results in [10] demonstrate the effectiveness of regularizing the Vendi score in guiding diffusion models toward more diverse sample distributions over multiple rounds of data generation. Another method, CADS [14] tends to improve sample diversity through the perturbation of conditional inputs, typically resulting in an increased Vendi score.

Although the mentioned diversity-score-based diffusion sampling methods apply smoothly to the unconditional sample generation without a varying input prompt, a scalable prompt-aware extension to prompt-guided diffusion models remains a challenge. As discussed in [10], one feasible framework extension to prompt-guided generation can leverage a pre-clustering of prompts to apply the Vendi diversity guidance separately within every prompt cluster. However, in successive queries to prompt-guided generative models, prompts often vary in subtle details, making their pre-clustering into a fixed number of hard clusters challenging. In addition, relying on small clusters of highly similar prompts could lead to limited diversity across prompt clusters, due to missing the partial similarities of the prompts in different clusters.

# Diversity Guidance in Latent Diffusion Models

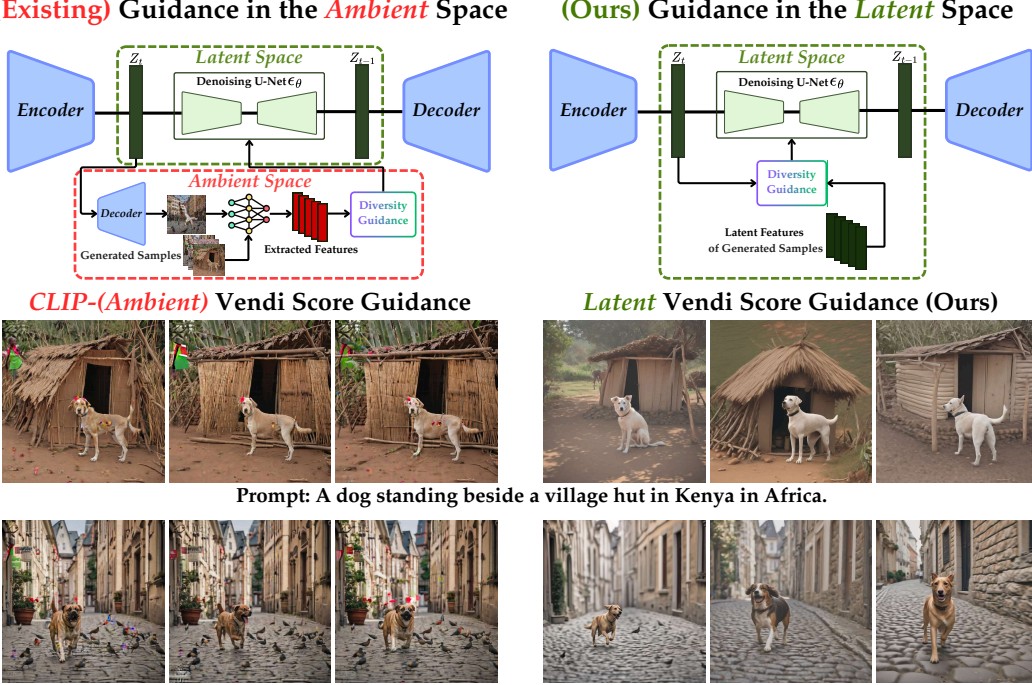

Figure 2: Comparison of latent entropy-based diversity guidance (ours) vs. ambient entropy diversity guidance in Latent Diffusion Models (LDMs). The experiment is performed with the SD-XL LDM.

In this work, we propose **S**calable **P**rompt-**A**ware **R**ény **K**ernel **E**ntropy Diversity Guidance (*SPARKE*) in diffusion models, applying the conditional entropy-based diversity score family [15] of the latent representation in the latent diffusion models (LDMs). As illustrated in Figure 1, SPARKE directly incorporates prompts into the diversity calculation without pre-clustering. This extension enables a more dynamic control over the diversity calculation with varying prompts, which can be interpreted as an adaptive kernel-based similarity evaluation without grouping the prompts into clusters. According to the conditional entropy guidance in SPARKE, we update the kernel matrix of generated data by taking the Hadamard product with the kernel matrix of input prompts. Applying an appropriate kernel function for text prompts, the updated similarity matrix will assign a higher weight to the pairwise interaction of samples with higher prompt similarity.

While the conditional entropy guidance offers flexibility by relaxing the requirement for explicit hard clustering of the prompts, it relies on computing the gradient of the matrix-based entropy of the kernel matrix, which will be computationally expensive in large-scale generation tasks. Since the entropy score is computed for an $n \times n$ kernel similarity matrix of $n$ samples, the entropy estimation of its eigenvalues would need an eigendecomposition, leading to $\mathcal{O}(n^3)$ complexity. This computational cost would be heavy in settings where thousands of samples need to be diversified across a large set of prompts. Thus, while conditioning diversity on prompts improves the adaptability of our proposed approach, we further need to develop a computationally scalable solution to extend the approach to a larger-scale generation of diverse data.

To improve scalability while preserving diversity-aware generation, we propose applying the order-2 Renyi matrix-based entropy, defined as the RKE Score [16], in the SPARKE approach. This RKE-based formulation replaces eigenvalue decomposition with a Frobenius norm-based entropy measure, which significantly reduces computational overhead. As a result, the complexity of entropy estimation reduces to $\mathcal{O}(n^2)$, while the gradient-based optimization in guided diffusion models can be performed even more efficiently in $\mathcal{O}(n)$ time due to the canceled terms with zero gradients. By integrating these computational improvements, we introduce SPARKE framework, which maintains the advantages of prompt-conditioned diversity guidance while making large-scale sample generation feasible.

**Latent Diffusion Model (LDM)**     **LDM + SPARKE Guidance (Ours)**

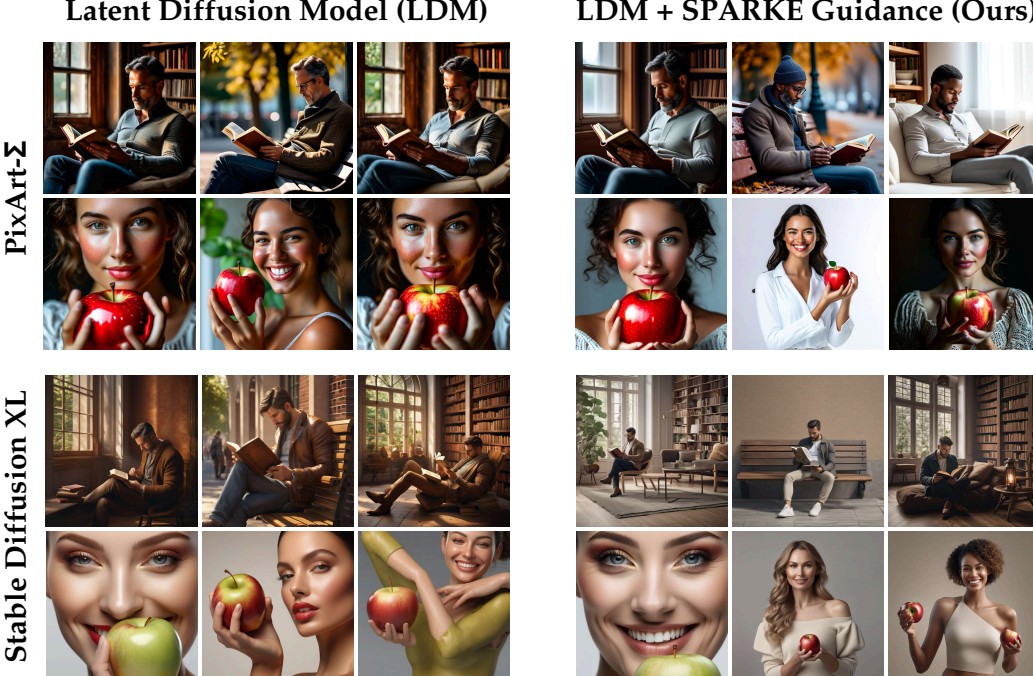

Prompts (row 1): 1. A man sitting quietly and reading a book, 2. A man reading book on a bench, 3. A man reading a book in a relaxed setting.

Prompts (row 2): 1. A young woman holding an apple, 2. A female holding an apple and smiling, 3. A woman showing apple with both hands.

Figure 3: Qualitative Comparison of samples generated by the base latent diffusion model (LDM), PixArt-$\Sigma$, and Stable Diffusion XL, vs. LDM guided via our proposed SPARKE guidance.

To evaluate the effectiveness of our approach, we perform several numerical experiments on standard text-to-image generation models. Our results indicate that Conditional RKE Score Guidance not only scales efficiently to a large number of prompts but also maintains high sample diversity while preserving fidelity. Compared to prior methods, our approach achieves a more balanced trade-off between diversity and computational efficiency, making it suitable for real-world generative modeling applications that require scalability. The following is a summary of the contributions of our work:

- We introduce the conditional entropy score guidance as a prompt-aware diversity promotion tool in sampling from diffusion models.

- We propose using the order-2 Renyi kernel entropy (RKE) score to reduce the computational complexity of the entropy-based diversity guidance.

- We propose the application of entropy guidance in the latent space of latent diffusion models, improving the efficiency and performance in the entropy-based diversity guidance (See Figure 2).

- We test our method on several state-of-the-art diffusion models (see Figures 3) and text-to-image benchmarks, indicating its ability to improve sample diversity and computational efficiency.

## 2   Related Works

**Standard and Latent Diffusion Models.** Diffusion generative models [1, 2, 17, 3] learns to reverse an iterative noising process, effectively estimating the gradient of the data log-density (the Stein score [18]) to generate new data samples. This approach has shown remarkable capabilities in synthesizing high-fidelity images [6, 19, 20, 7]. Despite the impressive results, a primary limitation of diffusion models was the substantial computational cost, particularly when operating directly in high-dimensional spaces like pixel space. To address this challenge, Latent Diffusion Models (LDMs) [4, 21] perform the forward and denoising processes in an encoded latent space, enabling high-quality images such as Stable Diffusion [4, 5] and video [22, 23, 8] synthesis at a large scale.

**Conditional Generation with Guidance.** The ability to control generative processes with specific conditions is increasingly crucial for practical applications, based on conditional inputs like text-

guided [20, 24, 25], class labels [6], style images [26, 27], or human motions [28], etc. Methods for conditional generation with guidance are categorized as either training-based or training-free. Training-based approaches either learn a time-dependent classifier that guides the noisy sample $x_t$ towards the condition $y$ [6, 24, 29, 25], or directly train the conditional denoising model $\epsilon_\theta(x_t, t, y)$ via few-shot adaptation [26, 4, 30]. In contrast, training-free guidance aims at zero-shot conditional generation by leveraging a pre-trained differentiable target predictor without requiring any training. This predictor can be a classifier, loss function, or energy function quantifying the alignment of a sample with the target condition [31, 32, 33, 34]. Our work can be included as a training-free guidance approach that applies conditional entropy scores guidance to enhance the diversity of samples.

**Quantifying Diversity and Novelty.** Diversity is quantified using both reference-based [35, 36] and reference-free metrics. Reference-free metrics include the Vendi Score [13, 37, 38], the RKE score [16] for unconditional models, and the Conditional-RKE [15] and Scendi [39] scores for conditional models. Also, the diversity metrics have been extended to online and distributed model selection tasks [40, 41, 42, 43]. For novelty, prior work [44, 45, 46] analyzes how generated samples differ from a reference model, with [45, 46] proposing a spectral method to measure the entropy of novel modes. [47, 48, 49, 50] also introduce kernel-based methods to compare and align two embeddings. In this work, we propose a novelty guidance approach that operates with respect to a reference dataset.

**Guidance for Improving Diversity.** A common strategy in diffusion-based generative modeling is the use of guidance mechanisms to balance quality and diversity [51, 52, 53]. For example, classifier-free guidance methods [52] considerably enhance prompt alignment and image quality but often compromise diversity due to overly deterministic conditioning. Several works have attempted to address this diversity challenge. To encourage diversity, [54] introduced a strategy that samples from the data manifold's low-density regions, however, their method operates directly in pixel space, posing challenges in adapting it effectively to latent diffusion frameworks. Another line of work is fine-tuning. In [55], the authors provide a finetuning method using Reinforcement Learning to improve the diversity of generated samples using a diversity reward function.

Recent works tackle this problem in the denoising phase. The CADS framework [14] shows that adding Gaussian noise to the conditioning signals during inference increases sample diversity. Particle Guidance (PG) [56] employs non-IID sampling from the joint distribution defined by a diffusion model combined with a potential function that maximizes pairwise dissimilarity across all samples, independent of semantic context. Similar to PG, ProCreate [57] uses DreamSim embeddings to find similar images via log energy and maximizes embedding-space distances. The concurrent method SPELL [58] adds repellency terms during sampling to prevent samples in a batch from being too close. These methods maximize across all samples without considering the prompts. However, these methods are prompt-agnostic, while ours dynamically conditions diversity guidance on the input.

The recent work [10] introduces contextualized Vendi Score Guidance (c-VSG), enhancing generative diversity during the denoising process using Vendi Score [13]. Their approach, however, requires the same prompts, restricting its applicability across diverse prompt scenarios. To tackle this issue, [15] uses Conditional-Vendi score guidance for a prompt-aware diversity guidance. On the other hand, our method leverages the RKE Score [16] to significantly enhance computational efficiency and sample complexity. Additionally, we propose prompt-aware guidance inspired by [15], enabling adaptive soft-clustering and effective conditioning on semantically distinct prompts. Moreover, unlike [10], which applies guidance based on latent encoded features of reference images, our strategy directly applies guidance within the diffusion model's latent space, saving computational costs and enabling a prompt-aware diversity improvement.

## 3 Preliminaries

### 3.1 Kernel function and Vendi diversity scores

Consider a sample $\mathbf{x} \in \mathcal{X}$ in the support set $\mathcal{X}$. A function $k : \mathcal{X} \times \mathcal{X} \to \mathbb{R}$ is called a kernel function, if for every integer $n \in \mathbb{N}$ and sample set $\{x_1, \ldots, x_n\} \in \mathcal{X}$, the following kernel similarity matrix $K \in \mathbb{R}^{n \times n}$ is positive semi-definite (PSD):

$$K_X = \begin{bmatrix} k(x_1, x_1) & \cdots & k(x_1, x_n) \\ \vdots & \ddots & \vdots \\ k(x_n, x_1) & \cdots & k(x_n, x_n) \end{bmatrix} \tag{1}$$

We assume that the kernel function is normalized, i.e., $k(x,x) = 1$ for every $x \in \mathcal{X}$. For a general (potentially unnormalized) kernel $k$, one can define its normalized counterpart as $\widetilde{k}(x,y) = k(x,y)/\sqrt{k(x,x)\,k(y,y)}$, which we apply whenever normalization is required. The Rényi Kernel Entropy (RKE) diversity score [16] is defined and analyzed as the inverse Frobenius norm-squared of the trace-normalized kernel matrix:

$$\text{RKE}(x_1,\ldots,x_n) := \exp\Big(H_2\big(\tfrac{1}{n}K_X\big)\Big) \; = \; \Big\|\tfrac{1}{n}K_X\Big\|_F^{-2} \tag{2}$$

where $\|\cdot\|_F$ denotes the Frobenius norm. As theoretically shown in [16], the RKE score can be regarded as a mode count of a mixture distribution with multiple modes.

To extend this entropy-based diversity scores to conditional (i.e., prompt-aware) diversity measurement for prompt-guided generative models, [15] propose the application of conditional kernel matrix entropy measures, resulting in the following definition of order-2 Conditional-RKE given the Hadamard product $\frac{1}{n}K_X \odot K_T$ of the output data $K_X$ and input prompt $K_T$ kernel matrices:

$$\text{Conditional-RKE}(x_1,\ldots,x_n;t_1,\ldots,t_n) := \frac{\big\|K_T\big\|_F^2}{\big\|K_X \odot K_T\big\|_F^2}, \tag{3}$$

Note that here the Hadamard product of the prompt $(t_1,\ldots t_n)$ kernel matrix $K_T$ and output $(x_1,\ldots x_n)$ kernel matrix $K_X$ is defined as the elementwise product of the matrices, which is guaranteed to be a PSD matrix by the Schur product theorem.

### 3.2 Conditional Latent Diffusion Models

Latent Diffusion Models (LDMs) [4] can achieve scalability and efficiency by performing the diffusion process within a compressed latent space, rather than directly in pixel space. A pre-trained autoencoder is used in LDMs, consisting of an encoder $\mathcal{E}$ and a decoder $\mathcal{D}$. The encoder $\mathcal{E}$ maps high-dimensional images $\boldsymbol{x} \in \mathbb{R}^{H \times W \times C}$ to a lower-dimensional latent representation $\boldsymbol{z}_0 = \mathcal{E}(\boldsymbol{x}) \in \mathbb{R}^{h \times w \times c}$, while the decoder $\mathcal{D}$ reconstructs the images $\tilde{\boldsymbol{x}} = \mathcal{D}(\boldsymbol{z}_0)$ from a denoised latent variable $\boldsymbol{z}_0$. The forward diffusion process applies in the latent space as $\boldsymbol{z}_t = \sqrt{\alpha_t}\boldsymbol{z}_0 + \sqrt{1-\alpha_t}\boldsymbol{\epsilon}_t$, where $\boldsymbol{\epsilon}_t \sim \mathcal{N}(\boldsymbol{0},\boldsymbol{I})$ and $\alpha_t \in$ are predefined noise schedule parameters. Then the conditional LDMs with conditions $\boldsymbol{y}$ are trained to predict the noise $\boldsymbol{\epsilon}_\theta(\boldsymbol{z}_t,t,\boldsymbol{y})$ at each time step $t$, also learn the score of $p_t(\boldsymbol{z_t}|\boldsymbol{y})$ [17, 3]:

$$\min_\theta \mathbb{E}_{\boldsymbol{z}_t,\boldsymbol{\epsilon}_t,t,\boldsymbol{y}} \left[\big\|\boldsymbol{\epsilon}_\theta(\boldsymbol{z}_t,t,\boldsymbol{y}) - \boldsymbol{\epsilon}_t\big\|_2^2\right] = \min_\theta \mathbb{E}_{\boldsymbol{z}_t,\boldsymbol{\epsilon}_t,t,\boldsymbol{y}} \left[\big\|\boldsymbol{\epsilon}_\theta(\boldsymbol{z}_t,t,\boldsymbol{y}) + \sqrt{1-\alpha_t}\nabla_{\boldsymbol{z}_t}\log p_t(\boldsymbol{z}_t|\boldsymbol{y})\big\|_2^2\right]. \tag{4}$$

## 4 Scalable Prompt-Aware Diversity Guidance in Diffusion Models

Utilizing the kernel-based entropy diversity scores and latent diffusion models (LDMs) described in Section 3.2, we introduce a scalable conditional entropy-based framework extending the recent Vendi score-based approach [10] in diversity-guided generative modeling to prompt-aware diversity enhancement that suits the conditional text-to-image models. Specifically, we first propose extending the order-1 Vendi score to the general matrix-based order-$\alpha$ Rényi entropy and subsequently to the conditional order-$\alpha$ Rényi entropy for achieving higher prompt-conditioned diversity. In addition, we demonstrate a computationally efficient special case in this family of diversity score functions by considering the order-2 entropy measures. This choice of entropy function leads to the RKE-guidance and Conditional-RKE guidance approaches for enhancing overall diversity and prompt-aware diversity in sample generation via diffusion models.

### 4.1 Diversity guidance via general order-$\alpha$ Renyi Entropy and RKE scores

The existing diversity score-based guidance approach, including the Vendi guidance in [10], optimizes the diversity score of the embedded version of the generated outputs, e.g. the CLIP embedding [59] of image data generated by the prompt-guided diffusion models. However, such a guidance process by considering an embedding on top of the diffusion model is computationally expensive and may not lead to semantically diverse samples. As discussed in [13, 37], computing the order-1 Vendi score and its gradients requires at least $\Omega(n^{2.367})$ computations for $n$ samples, and in practice involves

$O(n^3)$ computations for the eigen-decompositions of kernel matrices. This computational complexity limits practical window sizes of the existing Vendi guidance approach to a few hundred samples, i.e. the guidance function cannot be computed by a standard GPU processor for a sample size $n > 500$.

To reduce computational complexity, we extend the order-1 Vendi score guidance to the general order-$\alpha$ Rènyi kernel entropy, and adopt the specific case $\alpha = 2$, referred to as the RKE score [16]. This formulation significantly lowers computational cost and is defined as:

$$\mathcal{L}_{\text{RKE}}(\boldsymbol{z}^{(1)}, \dots, \boldsymbol{z}^{(n)}) = \left\| \tfrac{1}{n} K_Z \right\|_F^{-2} = \left( \frac{1}{n^2} \sum_{i=1}^n \sum_{j=1}^n k^2(\boldsymbol{z}^{(i)}, \boldsymbol{z}^{(j)}) \right)^{-1}, \tag{5}$$

with kernel matrix $(K_Z)_{ij} = k(\boldsymbol{z}^{(i)}, \boldsymbol{z}^{(j)})$ computed directly for the latent representations and $n$ denotes the total number of generated samples. Therefore, applying $\mathcal{L}_{\text{RKE}}$ as the guidance potential function reduces the computational complexity from $O(n^3)$ in order-1 Vendi score to $O(n^2)$, enabling significantly larger batch sizes during the guided sampling process and resulting in higher efficiency.

## 4.2 Diversity-Guided Sampling in Latent Diffusion Models

In contrast to previous work using order-1 Vendi score as the diversity guidance in ambient image space [10], we integrate efficient IRKE score diversity guidance directly into latent-space sampling. Standard LDMs employ classifier-free guidance (CFG) [52] for conditional sampling, iteratively denoising latents $\boldsymbol{z}_{t-1}$ from the noisy latents $\boldsymbol{z}_t$ at time step $t$ via reverse samplers:

$$\boldsymbol{z}_{t-1} \leftarrow \text{Sampler}(\boldsymbol{z}_t, \hat{\boldsymbol{\epsilon}}_\theta(\boldsymbol{z}_t, t, \boldsymbol{y})), \tag{6}$$

To efficiently promote diversity within the latent space, we optimize the Inverse-RKE (IRKE) score loss. Proposition 1 defines this loss and its gradient:

**Proposition 1.** *Let $Z = \{\boldsymbol{z}^{(1)}, \boldsymbol{z}^{(2)}, \dots, \boldsymbol{z}^{(n)}\}$ denote a set of $n$ generated data. Let kernel function $k : \mathcal{Z} \times \mathcal{Z} \to \mathbb{R}$ be symmetric and normalized, i.e. $k(\boldsymbol{z}, \boldsymbol{z}) = 1$ for every $\boldsymbol{z} \in \mathcal{Z}$. Then, we observe that the function $\mathcal{L}_{RKE}(\boldsymbol{z}^{(1)}, \dots, \boldsymbol{z}^{(n)})$ in Eq. (5) changes montonically with the Inverse-RKE function $\mathcal{L}_{IRKE}(\boldsymbol{z}^{(1)}, \dots, \boldsymbol{z}^{(n)}) = 1/\mathcal{L}_{RKE}(\boldsymbol{z}^{(1)}, \dots, \boldsymbol{z}^{(n)})$, whose gradient with respect to $\boldsymbol{z}^{(n)}$ is:*

$$\nabla_{\boldsymbol{z}^{(n)}} \mathcal{L}_{IRKE} \propto \sum_{i=1}^{n-1} k(\boldsymbol{z}^{(i)}, \boldsymbol{z}^{(n)}) \nabla_{\boldsymbol{z}^{(n)}} k(\boldsymbol{z}^{(i)}, \boldsymbol{z}^{(n)}). \tag{7}$$

The above Proposition 1 implies that an optimization objective maximizing $\mathcal{L}_{\text{RKE}}$ to promote diversity can be equivalently pursued by minimizing $\mathcal{L}_{\text{IRKE}}$, reducing the computational complexity from $O(n^2)$ to $O(n)$. Leveraging this computational efficacy, our approach directly updates the gradient of $\mathcal{L}_{\text{IRKE}}$ rather than computing the gradient of $\mathcal{L}_{\text{RKE}}$ to the latent $\boldsymbol{z}^{(n)}$, allowing for an efficient, explicit diversity-promoting update. Specifically, for Latent Diffusion Models (LDMs), we introduce the following diversity-guided sampling via Eq. (7) in each time step $t$:

$$\boldsymbol{z}_t \leftarrow \boldsymbol{z}_t - \eta \cdot \nabla_{\boldsymbol{z}^{(n)}} \mathcal{L}_{\text{IRKE}} \quad (\textbf{IRKE Diversity Guidance}) \tag{8}$$

where $\eta$ represents the guidance scale and $n$ denotes the number of previously generated samples.

## 4.3 Conditional Prompt-Aware Diversity Guidance

To explicitly capture prompt-conditioned diversity and focus on the computational efficiency of order-2, we introduce the Conditional RKE loss:

$$\mathcal{L}_{\text{Cond-RKE}}(\boldsymbol{z}^{(1)}, \dots, \boldsymbol{z}^{(n)}; \boldsymbol{y}^{(1)}, \dots, \boldsymbol{y}^{(n)}) := \frac{\left\| K_Y \right\|_F^2}{\left\| K_Y \odot K_Z \right\|_F^2}, \tag{9}$$

where $(K_Y)_{ij} = k_Y(\boldsymbol{y}^{(i)}, \boldsymbol{y}^{(j)})$ denotes the similarity among condition (e.g., prompts) embeddings. Similarly, we reduce the computational complexity from $O(n^2)$ to $O(n)$, by computing the gradient of Conditional Inverse-RKE (Cond-IRKE) defined in Proposition 2:

**Proposition 2.** *Let $Z$ denote a set of $n$ generated data, and $Y = \{\boldsymbol{y}^{(1)}, \boldsymbol{y}^{(2)}, \dots, \boldsymbol{y}^{(n)}\}$ is the set of corresponding conditions. Let the kernel function $k$ be symmetric and normalized. Then, we observe that the function $\mathcal{L}_{Cond-RKE}(\boldsymbol{z}^{(1)}, \dots, \boldsymbol{z}^{(n)}; \boldsymbol{y}^{(1)}, \dots, \boldsymbol{y}^{(n)})$ in Eq. (9) changes montonically with*

**Algorithm 1:** Scalable Prompt-Aware Latent IRKE Diversity Guidance

---

**Input:** Latents $\{z^{(1)}, \ldots, z^{(n-1)}\}$, prompt features $\{y^{(1)}, \ldots, y^{(n-1)}\}$, kernel functions $k_Z$, $k_Y$, denoising model $\epsilon_\theta$, diffusion reverse sampler, guidance scale $w$, diversity guidance scale $\eta$, decoder $\mathcal{D}$

**Output:** Diverse generated samples $\{\tilde{x}^{(n)}\}$ with prompt features $y^{(n)}$

1 **for** $t = T$ **to** $1$ **do**

2      Compute CFG guided noise: $\hat{\epsilon}_\theta^{(n)} = (1 + w) \cdot \epsilon_\theta(z_t^{(n)}, t, y^{(n)}) - w \cdot \epsilon_\theta(z_t^{(n)}, t)$ ;

3      Perform one-step denoising: $z_{t-1}^{(n)} \leftarrow \text{Sampler}(z_t^{(n)}, \hat{\epsilon}_\theta^{(n)})$ ;

4      Compute $K_Z, K_Y$, and $\mathcal{L}_{\text{Cond-IRKE}} = \frac{1}{n^2} \| K_Z \odot K_Y \|_F^2$ ;

5      Compute the gradients $g^{(n)} = \nabla_{z^{(n-1)}} \mathcal{L}_{\text{Cond-IRKE}}$ ;

6      Update latents: $z_{t-1}^{(n)} \leftarrow z_{t-1}^{(n)} - \eta \cdot g^{(n)}$ ;

7 Decode final samples: $\tilde{x}^{(n)} = \mathcal{D}(z_0^{(n)})$

---

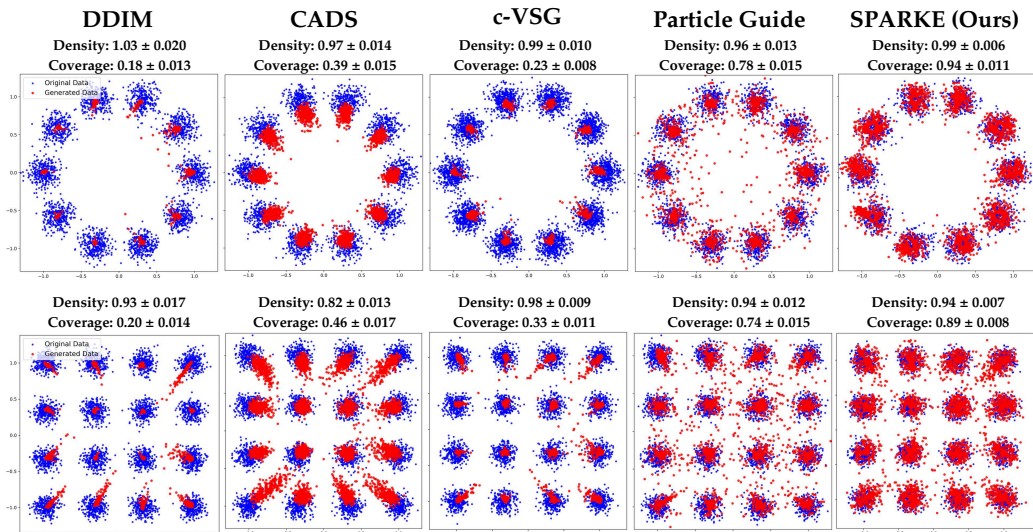

Figure 4: Comparison of SPARKE (Conditional RKE) Guidance with baselines on 2D GMMs.

the Conditional Inverse-RKE function $\mathcal{L}_{Cond\text{-}IRKE}(z^{(1)}, \ldots, z^{(n)}; y^{(1)}, \ldots, y^{(n)}) = \| \widetilde{K}_Z \odot \widetilde{K}_Y \|_F^2$, whose gradient with respect to $z^{(n)}$ is:

$$\nabla_{z^{(n)}} \mathcal{L}_{Cond\text{-}IRKE} \propto \sum_{i=1}^{n-1} k_Z(z^{(i)}, z^{(n)}) k_Y(y^{(i)}, y^{(n)})^2 \nabla_{z^{(n)}} k_Z(z^{(i)}, z^{(n)}). \tag{10}$$

Then we propose the following prompt-aware diversity-guided sampling via Eq. (10) in time step $t$:

$$z_t \leftarrow z_t - \eta \cdot \nabla_{z^{(n)}} \mathcal{L}_{\text{Cond-IRKE}} \qquad (\textbf{Cond-IRKE Diversity Guidance}) \tag{11}$$

Algorithm 1 provides an explicit outline of this procedure. This Cond-IRKE guidance explicitly aligns the diversity of generated samples with respective prompts, while significantly maintaining the computational efficiency in gradient calculation. In Appendix B.3, we provide a theoretical interpretation based on a stochastic differential equation.

## 5 Numerical Results

We evaluated the performance of the SPARKE framework on various conditional diffusion models, where our results support that SPARKE can boost output diversity without compromising fidelity scores considerably. For the complete set of our numerical results, we refer to the Appendix D.

**Baselines.** We compare our method with CADS [14], Particle Guidance [56], and Contextualized Vendi Score Guidance (c-VSG) [10]. In experiments without a reference, we use VSG [10].

Table 1: Quantitative comparison of guidance methods on Stable Diffusion 2.1.

| Method | CLIPScore ↑ | KD×10² ↓ | Cond-Vendi Score ↑ | Vendi Score ↑ | In-batch Sim.×10² ↓ |
|---|---|---|---|---|---|
| SD (No Guidance) [60] | 31.20 | 52.37 | 26.54 | 369.37 | 80.36 |
| c-VSG [10] | 28.75 | 56.25 | 27.91 | 376.48 | 79.82 |
| CADS [14] | 29.89 | 55.08 | 28.73 | 380.08 | 79.44 |
| **latent RKE Guidance (Ours)** | 30.18 | 55.37 | 29.88 | 387.59 | 79.01 |
| **SPARKE: latent Cond-RKE (Ours)** | 30.96 | 53.15 | 32.57 | 405.51 | 75.68 |

## Stable Diffusion 2.1

| Prompt-Unaware Diversity Guidance (RKE score) | SPARKE: Prompt-Aware Diversity Guidance (Conditional RKE score) |
|---|---|
| In-batch Sim.: 79.01 | In-batch Sim. : 75.68 |

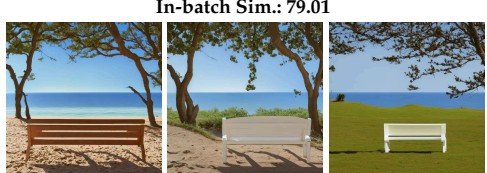 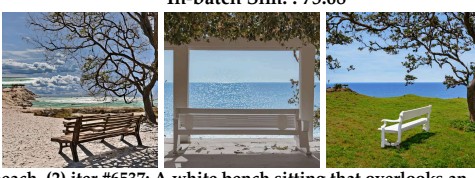

**Prompts: (1) iter #6219: This is an image of a bench overlooking a beach, (2) iter #6537: A white bench sitting that overlooks an ocean view, (3) iter #9353.A white wooden bench sitting on top of a green hill next to the ocean.**

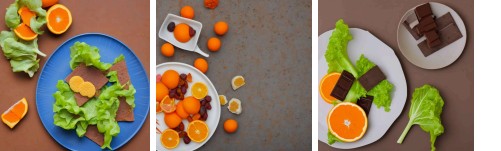 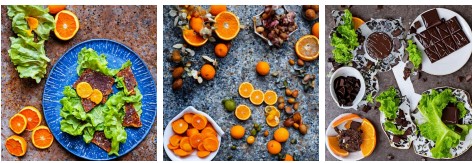

**Prompts: (1) iter #7927: a blue plate with an orange a cracker some lettuce and a twist bar, (2) iter #8354: A dish contains an orange and snacks., (3) iter #9287: An orange, chocolate, cracker, and piece of lettuce on a plate**

Figure 5: Comparison of SPARKE prompt-aware diversity guidance via conditional-RKE score vs. diversity-unaware diversity guidance using the RKE score on SD 2.1 text-to-image generation.

**Models.** We used Stable Diffusion [60] and the larger-scale Stable Diffusion XL (SDXL) [5], and PixArt-$\Sigma$ [61] in our experiments on text-to-image generation. We also provide additional experimental results on different SOTAs in the Appendix.

**Evaluation.** We compared our method with baselines in terms of output diversity and fidelity. Fidelity was measured using CLIPScore [62], and KID [63] to evaluate prompt quality and consistency, and Density [36] where a reference dataset was available. For diversity, we used Vendi [13] and Conditional-Vendi [15] scores, and we used Coverage [36] when a reference dataset existed. We measured diversity within each prompt cluster using the in-batch similarity score [56], calculated as the average pairwise cosine similarity of image features in a batch.

**Synthetic Datasets.** We compare our method on 2D Gaussian mixture benchmarks [64], training a diffusion model with the DDIM sampler [65] and fine-tuning hyperparameters (see Appendix C). Figure 4 shows that CADS covered the modes' support but with a slight distribution shift, while c-VSG better adhered to the modes at the cost of lower diversity. PG and SPARKE both covered the modes well, with SPARKE producing fewer noisy samples due to its clustering-based structure.

**Comparison of Entropy Guidance Effects in Latent and Ambient Spaces.** We compared latent-space guidance (SPARKE) with CLIP-embedded ambient-space guidance from c-VSG [10] using modified GeoDE categories [66]. As qualitatively shown in Figure 2, CLIP-ambient led to visual artifacts and offered less improvement in output visual diversity, while latent guidance produced more semantically diverse outputs. We also provided quantitative results in Table 5, which show that latent guidance achieves higher diversity. Also, latent guidance requires significantly less GPU memory ($\approx$20GB vs. $\approx$35GB), offering substantially better computational efficiency.

**Comparison of Prompt-Aware Diversity Guidance vs. Unconditional Guidance.** We generated images for 10,000 MS-COCO prompts [67] using Stable Diffusion 2.1, comparing latent RKE conditional guidance (prompt-aware) against unconditional RKE guidance (prompt-unaware). Figure 5 shows unconditional diversity guidance becomes less effective over time, while SPARKE's prompt-aware approach remained effective throughout all 40k iterations. Table 1 confirms that prompt-aware

Table 2: Comparison of diversity and fidelity metrics across different models and methods.

| Method | CLIPScore ↑ | KD×10² ↓ | Cond-Vendi Score ↑ | Vendi Score ↑ | In-batch Sim.×10² ↓ |
|---|---|---|---|---|---|
| SDXL | 31.17 | 66.37 | 27.54 | 309.54 | 81.67 |
| **SDXL + SPARKE** | 30.47 | 62.62 | 31.17 | 313.80 | 77.75 |
| PixArt-Σ | 31.01 | 65.37 | 26.45 | 307.36 | 83.84 |
| **PixArt-Σ + SPARKE** | 30.66 | 63.86 | 32.14 | 322.25 | 78.81 |

### Novelty Guidance Using SPARKE

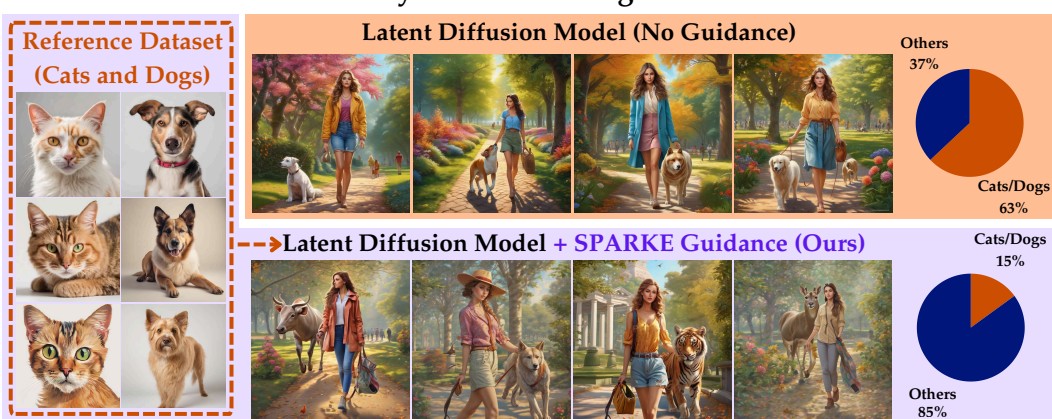

Figure 6: Novelty guidance with SPARKE. Using the prompt "A young lady walking with an animal in the park," samples are generated with respect to a reference set of cat and dog images, resulting in more novel samples compared to the base model.

methods achieve higher diversity while preserving image quality and prompt alignment. Also, Table 2 provides the quantitative results for SD-XL and PixArt-Σ.

**Novelty Guidance Using SPARKE.** We evaluated novelty guidance by using the SPARKE method to guide an SD-XL model with a reference dataset of cat and dog images. As shown in Figure 6, this approach successfully reduced the generation of cats and dogs, yielding more novel samples with respect to the reference dataset and resulting in a more balanced distribution of other animals.

**Computational Efficiency Comparison.** We evaluated SPARKE's computational efficiency by comparing its runtime and peak memory with the baselines over 1000 text-to-image generations using a 50-step diffusion process on an NVIDIA RTX 4090. The results, as shown in Table 3, indicate that SPARKE requires significantly lower runtime and less memory (as it functions in the latent space).

Table 3: Comparison of runtime and GPU memory usage for different guidance methods.

| Model + Guidance Method | Runtime per sample (s) | GPU Memory Peak (GB) |
|---|---|---|
| Stable Diffusion v1.5 | $1.620 \pm 0.265$ | $3.178 \pm 0.001$ |
| SD-1.5 + SPARKE (Ours) | $1.752 \pm 0.287$ | $3.230 \pm 0.025$ |
| SD-1.5 + c-VSG | $3.079 \pm 0.296$ | $8.665 \pm 0.116$ |
| SD-1.5 + Particle Guide (Pixel) | $4.912 \pm 0.312$ | $7.531 \pm 0.032$ |
| SD-1.5 + Particle Guide (DINOv2) | $9.102 \pm 0.271$ | $20.133 \pm 0.038$ |

## 6 Conclusion and Limitations

In this work, we proposed the prompt-aware SPARKE diversity guidance approach for prompt-based diffusion models. The SPARKE method aims to improve the diversity of output data conditioned on the prompt, so that the diversity guidance process takes into account the similarity of the prompts. We also proposed the application of RKE and Conditional-RKE scores in the latent space of LDMs to boost the scalability in the SPARKE method. Our numerical results of applying SPARKE to the stable-diffusion LDMs indicate the method's qualitative and quantitative improvement of variety in generated samples. A limitation of our numerical evaluation is its primary focus on image generation diffusion models. Future exploration can extend SPARKE's application to other modalities, such as video and text diffusion models. Furthermore, combining SPARKE's entropy guidance with other diversity-enhancement techniques for diffusion models presents another interesting future direction.

## Acknowledgments

The work of Farzan Farnia is partially supported by a grant from the Research Grants Council of the Hong Kong Special Administrative Region, China, Project 14209920, and is partially supported by CUHK Direct Research Grants with CUHK Project No. 4055164 and 4937054. The work of Amin Gohari is supported by CUHK Direct Research Grants with CUHK Project No. 4055270. The work is also supported by a grant under 1+1+1 CUHK-CUHK(SZ)-GDSTC Joint Collaboration Fund. Finally, the authors would like to sincerely thank the anonymous reviewers for their insightful feedback and constructive suggestions.

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

# A Preliminaries on Diffusion Models

## A.1 Denoising Diffusion Probabilistic Models (DDPMs)

Denoising diffusion probabilistic models (DDPMs) [1, 2] define a generative process by reversing a fixed Markovian forward diffusion that progressively adds Gaussian noise to data. Given a data sample $\boldsymbol{x}_0 \sim p_0(\boldsymbol{x}_0)$, a time step $t \in [T] \triangleq \{1, \ldots, T\}$, the forward process gradually adds noise to construct a noisy data point $\boldsymbol{x}_t = \sqrt{\alpha_t}\boldsymbol{x}_0 + \sqrt{1 - \alpha_t}\boldsymbol{\epsilon}_t$, where $\boldsymbol{\epsilon}_t \sim \mathcal{N}(\boldsymbol{0}, \boldsymbol{I})$ is standard Gaussian noise and $\alpha_t \in [0, 1]$ monotonically decreases with time step $t$ to control the noise level. The diffusion model $\boldsymbol{\epsilon}_\theta : \mathcal{X} \times [T] \mapsto \mathcal{X}$ is trained to predict the noise $\boldsymbol{\epsilon}_t$ at each time step $t$, also learn the score of $p_t(\boldsymbol{x_t})$ [17, 3]:

$$\min_\theta \mathbb{E}_{\boldsymbol{x}_t, \boldsymbol{\epsilon}_t, t} \left[ \left\| \boldsymbol{\epsilon}_\theta(\boldsymbol{x}_t, t) - \boldsymbol{\epsilon}_t \right\|_2^2 \right] = \min_\theta \mathbb{E}_{\boldsymbol{x}_t, \boldsymbol{\epsilon}_t, t} \left[ \left\| \boldsymbol{\epsilon}_\theta(\boldsymbol{x}_t, t) + \sqrt{1 - \alpha_t}\nabla_{\boldsymbol{x}_t} \log p_t(\boldsymbol{x}_t) \right\|_2^2 \right], \quad (12)$$

The reverse process obtained by $\boldsymbol{x}_{t-1} \sim p_{t-1|t}(\boldsymbol{x}_{t-1}|\boldsymbol{x}_t)$ is not directly computable in practice, multiple efficient reverse samplers are proposed [65, 68, 69]. In the commonly-used DDIM [65] sampler, we sample $\boldsymbol{x}_{t-1}$ by:

$$\boldsymbol{x}_{t-1} = \sqrt{\alpha_{t-1}}\tilde{\boldsymbol{x}}_{0|t} + \sqrt{1 - \alpha_{t-1} - \sigma_t^2}\frac{\boldsymbol{x}_t - \sqrt{\alpha_t}\tilde{\boldsymbol{x}}_{0|t}}{\sqrt{1 - \alpha_t}} + \sigma_t\boldsymbol{\epsilon}_t, \quad (13)$$

where $\sigma_t$ is the DDIM parameter, and the clean sample $\tilde{\boldsymbol{x}}_{0|t}$ given $\boldsymbol{x}_t$ is estimated according to Tweedie's Formula [70, 71]:

$$\tilde{\boldsymbol{x}}_{0|t} = \frac{\boldsymbol{x}_t - \sqrt{1 - \alpha_t}\boldsymbol{\epsilon}_\theta(\boldsymbol{x}_t, t)}{\sqrt{\alpha_t}}. \quad (14)$$

## A.2 Conditional Generation via Guidance.

In conditional generation tasks like text-to-image synthesis, diffusion models learn to approximate the conditional distribution $p(\boldsymbol{x}_0|\boldsymbol{y})$ given conditions $\boldsymbol{y}$ (e.g., text prompts). From the view of score functions [17, 3], we denote the conditional score as:

$$\underbrace{\nabla_{\boldsymbol{x}_t} \log p_t(\boldsymbol{x}_t|\boldsymbol{y})}_{\text{Conditional Score}} = \underbrace{\nabla_{\boldsymbol{x}_t} \log p_t(\boldsymbol{x}_t)}_{\text{Unconditional Score}} + \underbrace{\nabla_{\boldsymbol{x}_t} \log p_t(\boldsymbol{y}|\boldsymbol{x}_t)}_{\text{Guidance Score}}, \quad (15)$$

The strategies of conditional generation via guidance can be roughly divided into two categories: Training-based methods and training-free methods.

**Training-based Guidance.** Training-based guidance methods include several strategies. One approach, Classifier-Guidance, initially proposed in [3, 6], requires training an additional time-dependent classifier to estimate the guidance score $f(\boldsymbol{x}_t, t) \triangleq \mathbb{E}_{\boldsymbol{x}_0 \sim p_{0|t}(\cdot|\boldsymbol{x}_t)}f(\boldsymbol{x}_0) \approx f(\boldsymbol{x}_0)$. Alternatively, other training-based techniques involve few-shot fine-tuning of base models or the use of adapters [4, 30] to achieve conditional control.

A distinct yet training-based approach is Classifier-Free Guidance (CFG), introduced by [52]. Unlike methods requiring a separate classifier, CFG integrates the condition $\boldsymbol{y}$ as a direct input to the conditional denoising network $\boldsymbol{\epsilon}_\theta(\boldsymbol{x}_t, t, \boldsymbol{y})$. It is enabled by a joint training procedure where the model also learns to make unconditional predictions by randomly dropping the condition $\boldsymbol{y}$ with a specific probability during each training iteration. During inference, CFG estimates the conditional noise prediction as follows:

$$\hat{\boldsymbol{\epsilon}}_\theta(\boldsymbol{x}_t, t, \boldsymbol{y}) = (1 + w) \cdot \boldsymbol{\epsilon}_\theta(\boldsymbol{x}_t, t, \boldsymbol{y}) - w \cdot \boldsymbol{\epsilon}_\theta(\boldsymbol{x}_t, t), \quad (16)$$

where the guidance scale $w > 0$ adjusts the strength of conditional guidance. These various training-based guidance methods have demonstrated considerable effectiveness with the availability of training resources.

**Training-free Guidance.** An alternative category for conditional guidance is training-free guidance. Instead of requiring additional training, these methods directly introduce a time-independent conditional predictor $f$ on the estimated clean data $f(\tilde{\boldsymbol{x}}_{0|t}, \boldsymbol{y})$, which can be a classifier, loss function, or energy function quantifying the alignment of a generated sample with the target condition [32, 33, 34, 72]. Then the estimation of the guidance score via training-free approaches is:

$$\nabla_{\boldsymbol{x}_t} \log p_t(\boldsymbol{y}|\boldsymbol{x}_t) := -\nabla_{\boldsymbol{x}_t} f\left(\tilde{\boldsymbol{x}}_{0|t}, \mathbf{y}\right), \quad (17)$$

where the clean sample for the predictor is estimated by Tweedie's Formula [70, 71]:

$$\tilde{\boldsymbol{x}}_{0|t} = \frac{\boldsymbol{x}_t - \sqrt{1 - \alpha_t}\boldsymbol{\epsilon}_\theta(\boldsymbol{x}_t, t)}{\sqrt{\alpha_t}}. \tag{18}$$

# B  Proofs

## B.1  Proof of Proposition 1

*Proof.* Let $K_{\mathcal{Z}}$ be the kernel matrix with entries $(K_Z)_{ij} = k(\boldsymbol{z}^{(i)}, \boldsymbol{z}^{(j)})$. The RKE loss is given by:

$$\mathcal{L}_{\text{RKE}}(\boldsymbol{z}^{(1)}, \ldots, \boldsymbol{z}^{(n)}) = \frac{1}{\|\widetilde{K}_Z\|_F^2} = \frac{\text{Tr}(K_Z)^2}{\sum_{i=1}^n \sum_{j=1}^n k(\boldsymbol{z}^{(i)}, \boldsymbol{z}^{(j)})^2}, \tag{19}$$

The Inverse-RKE function $\mathcal{L}_{\text{IRKE}}$ is defined by:

$$\mathcal{L}_{\text{IRKE}}(\boldsymbol{z}^{(1)}, \ldots, \boldsymbol{z}^{(n)}) = \frac{1}{\mathcal{L}_{\text{RKE}}(\boldsymbol{z}^{(1)}, \ldots, \boldsymbol{z}^{(n)})} = \frac{\sum_{i=1}^n \sum_{j=1}^n k(\boldsymbol{z}^{(i)}, \boldsymbol{z}^{(j)})^2}{\text{Tr}(K_Z)^2}, \tag{20}$$

Given that the kernel function $k$ is normalized, i.e., $k(\boldsymbol{z}, \boldsymbol{z}) = 1$ for all $\boldsymbol{z} \in Z$, the trace of the kernel matrix $K_Z$ is:

$$\text{Tr}(K_Z) = \sum_{i=1}^n k(\boldsymbol{z}^{(i)}, \boldsymbol{z}^{(i)}) = \sum_{i=1}^n 1 = n,$$

Substituting this into the expression for $\mathcal{L}_{\text{IRKE}}(\boldsymbol{z}^{(1)}, \ldots, \boldsymbol{z}^{(n)})$ in Eq. (20):

$$\mathcal{L}_{\text{IRKE}}(\boldsymbol{z}^{(1)}, \ldots, \boldsymbol{z}^{(n)}) = \frac{1}{n^2} \sum_{i=1}^n \sum_{j=1}^n k(\boldsymbol{z}^{(i)}, \boldsymbol{z}^{(j)})^2, \tag{21}$$

Now, we want to compute the gradient of $\mathcal{L}_{\text{IRKE}}(\boldsymbol{z}^{(1)}, \ldots, \boldsymbol{z}^{(n)})$ with respect to $\boldsymbol{z}^{(n)}$:

$$\nabla_{\boldsymbol{z}^{(n)}} \mathcal{L}_{\text{IRKE}}(\boldsymbol{z}^{(1)}, \ldots, \boldsymbol{z}^{(n)}) = \nabla_{\boldsymbol{z}^{(n)}} \left( \frac{1}{n^2} \sum_{i=1}^n \sum_{j=1}^n k(\boldsymbol{z}^{(i)}, \boldsymbol{z}^{(j)})^2 \right), \tag{22}$$

Since $1/n^2$ is a constant with respect to $\boldsymbol{z}^{(n)}$:

$$\nabla_{\boldsymbol{z}^{(n)}} \mathcal{L}_{\text{IRKE}}(\boldsymbol{z}^{(1)}, \ldots, \boldsymbol{z}^{(n)}) = \frac{1}{n^2} \nabla_{\boldsymbol{z}^{(n)}} \left( \sum_{i=1}^n \sum_{j=1}^n k(\boldsymbol{z}^{(i)}, \boldsymbol{z}^{(j)})^2 \right), \tag{23}$$

Let $S = \sum_{i=1}^n \sum_{j=1}^n k(\boldsymbol{z}^{(i)}, \boldsymbol{z}^{(j)})^2$. We need to find $\nabla_{\boldsymbol{z}^{(n)}} S$. The terms in the sum $S$ that involve $\boldsymbol{z}^{(n)}$ are those where $i = n$ or $j = n$ (or both). We can split the sum into:

$$S = \sum_{i=1}^{n-1} \sum_{j=1}^{n-1} k(\boldsymbol{z}^{(i)}, \boldsymbol{z}^{(j)})^2 \quad \text{(Terms not involving } \boldsymbol{z}^{(n)})$$

$$+ \sum_{i=1}^{n-1} k(\boldsymbol{z}^{(i)}, \boldsymbol{z}^{(n)})^2 \quad (j = n, i \neq n)$$

$$+ \sum_{j=1}^{n-1} k(\boldsymbol{z}^{(n)}, \boldsymbol{z}^{(j)})^2 \quad (i = n, j \neq n)$$

$$+ k(\boldsymbol{z}^{(n)}, \boldsymbol{z}^{(n)})^2, \quad (i = n, j = n)$$

Using the symmetry of the kernel, $k(\boldsymbol{z}^{(n)}, \boldsymbol{z}^{(j)}) = k(\boldsymbol{z}^{(j)}, \boldsymbol{z}^{(n)})$, the second and third sums are identical:

$$S = \sum_{i=1}^{n-1} \sum_{j=1}^{n-1} k(\boldsymbol{z}^{(i)}, \boldsymbol{z}^{(j)})^2 + 2 \sum_{i=1}^{n-1} k(\boldsymbol{z}^{(i)}, \boldsymbol{z}^{(n)})^2 + k(\boldsymbol{z}^{(n)}, \boldsymbol{z}^{(n)})^2, \tag{24}$$

Now, we compute the gradient of $S$ with respect to $\boldsymbol{z}^{(n)}$:

$$\nabla_{\boldsymbol{z}^{(n)}} S = \nabla_{\boldsymbol{z}^{(n)}} \left( \sum_{i=1}^{n-1} \sum_{j=1}^{n-1} k(\boldsymbol{z}^{(i)}, \boldsymbol{z}^{(j)})^2 \right) + \nabla_{\boldsymbol{z}^{(n)}} \left( 2 \sum_{i=1}^{n-1} k(\boldsymbol{z}^{(i)}, \boldsymbol{z}^{(n)})^2 \right) + \nabla_{\boldsymbol{z}^{(n)}} \left( k(\boldsymbol{z}^{(n)}, \boldsymbol{z}^{(n)})^2 \right),$$

(25)

The first term is zero because it does not depend on $\boldsymbol{z}^{(n)}$. For the second term:

$$\begin{aligned} \nabla_{\boldsymbol{z}^{(n)}} \left( 2 \sum_{i=1}^{n-1} k(\boldsymbol{z}^{(i)}, \boldsymbol{z}^{(n)})^2 \right) &= 2 \sum_{i=1}^{n-1} \nabla_{\boldsymbol{z}^{(n)}} \left( k(\boldsymbol{z}^{(i)}, \boldsymbol{z}^{(n)})^2 \right) \\ &= 2 \sum_{i=1}^{n-1} 2 k(\boldsymbol{z}^{(i)}, \boldsymbol{z}^{(n)}) \nabla_{\boldsymbol{z}^{(n)}} k(\boldsymbol{z}^{(i)}, \boldsymbol{z}^{(n)}) \\ &= 4 \sum_{i=1}^{n-1} k(\boldsymbol{z}^{(i)}, \boldsymbol{z}^{(n)}) \nabla_{\boldsymbol{z}^{(n)}} k(\boldsymbol{z}^{(i)}, \boldsymbol{z}^{(n)}), \end{aligned}$$

(26)

For the third term, since $k(\boldsymbol{z}, \boldsymbol{z}) = 1$ (normalized kernel), this term is constant:

$$\nabla_{\boldsymbol{z}^{(n)}} \left( k(\boldsymbol{z}^{(n)}, \boldsymbol{z}^{(n)})^2 \right) = \nabla_{\boldsymbol{z}^{(n)}} (1^2) = \nabla_{\boldsymbol{z}^{(n)}} (1) = \boldsymbol{0},$$

(27)

According to Eq. (26) and Eq. (27), we have the gradeitn for $S$:

$$\nabla_{\boldsymbol{z}^{(n)}} S = 4 \sum_{i=1}^{n-1} k(\boldsymbol{z}^{(i)}, \boldsymbol{z}^{(n)}) \nabla_{\boldsymbol{z}^{(n)}} k(\boldsymbol{z}^{(i)}, \boldsymbol{z}^{(n)}),$$

(28)

Substituting this back into the gradient of $\mathcal{L}_{\text{IRKE}}(\boldsymbol{z}^{(1)}, \ldots, \boldsymbol{z}^{(n)})$ in Eq. (23):

$$\begin{aligned} \nabla_{\boldsymbol{z}^{(n)}} \mathcal{L}_{\text{IRKE}}(\boldsymbol{z}^{(1)}, \ldots, \boldsymbol{z}^{(n)}) &= \frac{1}{n^2} \left( 4 \sum_{i=1}^{n-1} k(\boldsymbol{z}^{(i)}, \boldsymbol{z}^{(n)}) \nabla_{\boldsymbol{z}^{(n)}} k(\boldsymbol{z}^{(i)}, \boldsymbol{z}^{(n)}) \right) \\ &= \frac{4}{n^2} \sum_{i=1}^{n-1} k(\boldsymbol{z}^{(i)}, \boldsymbol{z}^{(n)}) \nabla_{\boldsymbol{z}^{(n)}} k(\boldsymbol{z}^{(i)}, \boldsymbol{z}^{(n)}), \end{aligned}$$

(29)

Since $4/n^2$ is a positive constant (for $n \geq 1$), we have the following:

$$\nabla_{\boldsymbol{z}^{(n)}} \mathcal{L}_{\text{IRKE}} \propto \sum_{i=1}^{n-1} k(\boldsymbol{z}^{(i)}, \boldsymbol{z}^{(n)}) \nabla_{\boldsymbol{z}^{(n)}} k(\boldsymbol{z}^{(i)}, \boldsymbol{z}^{(n)}).$$

(30)

This proves the form of the gradient.

Regarding the monotonic relationship $\mathcal{L}_{\text{IRKE}} = 1/\mathcal{L}_{\text{RKE}}$, $\mathcal{L}_{\text{RKE}}$ is always positive ($\text{Tr}(K_Z)^2 = n^2 > 0$ and the denominator $\sum k_{ij}^2 \geq 0$; for it to be non-zero, not all kernel values can be zero), then $\mathcal{L}_{\text{RKE}}^2 > 0$. Therefore, $\frac{d\mathcal{L}_{\text{IRKE}}}{d\mathcal{L}_{\text{RKE}}} = -\frac{1}{\mathcal{L}_{\text{RKE}}^2} < 0$, which means that $\mathcal{L}_{\text{IRKE}}$ is a strictly monotonically decreasing function of $\mathcal{L}_{\text{RKE}}$. This completes the proof of Proposition 1. $\square$

## B.2 Proof of Proposition 2

*Proof.* Let $K_Z$ be the kernel matrix for the data $Z$ with entries $(K_Z)_{ij} = k_Z(\boldsymbol{z}^{(i)}, \boldsymbol{z}^{(j)})$. Let $K_Y$ be the kernel matrix for the conditions $Y$ with entries $(K_Y)_{ij} = k_Y(\boldsymbol{y}^{(i)}, \boldsymbol{y}^{(j)})$. The normalized kernel matrices are $\widetilde{K}_Z = K_Z/\text{Tr}(K_Z)$ and $\widetilde{K}_Y = K_Y/\text{Tr}(K_Y)$.
Given that the kernel functions $k_Z$ and $k_Y$ are normalized, their traces are:

$$\text{Tr}(K_Z) = \sum_{i=1}^{n} k_Z(\boldsymbol{z}^{(i)}, \boldsymbol{z}^{(i)}) = \sum_{i=1}^{n} 1 = n$$

$$\text{Tr}(K_Y) = \sum_{i=1}^{n} k_Y(\boldsymbol{y}^{(i)}, \boldsymbol{y}^{(i)}) = \sum_{i=1}^{n} 1 = n$$

So $\widetilde{K}_Z = K_Z/n$ and $\widetilde{K}_Y = K_Y/n$.

The Conditional Inverse-RKE function is defined as:

$$\mathcal{L}_{\text{Cond-IRKE}}(\boldsymbol{z}^{(1)}, \ldots, \boldsymbol{z}^{(n)}; \boldsymbol{y}^{(1)}, \ldots, \boldsymbol{y}^{(n)}) = \|\widetilde{K}_Z \odot \widetilde{K}_Y\|_F^2, \tag{31}$$

where $\odot$ denotes the Hadamard (element-wise) product. The entries of $\widetilde{K}_Z \odot \widetilde{K}_Y$ are:

$$\begin{aligned}
(\widetilde{K}_Z \odot \widetilde{K}_Y)_{ij} &= (\widetilde{K}_Z)_{ij}(\widetilde{K}_Y)_{ij} \\
&= \frac{1}{n}k_Z(\boldsymbol{z}^{(i)}, \boldsymbol{z}^{(j)}) \cdot \frac{1}{n}k_Y(\boldsymbol{y}^{(i)}, \boldsymbol{y}^{(j)}) \\
&= \frac{1}{n^2}k_Z(\boldsymbol{z}^{(i)}, \boldsymbol{z}^{(j)})k_Y(\boldsymbol{y}^{(i)}, \boldsymbol{y}^{(j)}),
\end{aligned} \tag{32}$$

The squared Frobenius norm is the sum of the squares of its elements, then we have the following according to Eq. (31):

$$\begin{aligned}
\mathcal{L}_{\text{Cond-IRKE}}(Z; Y) &= \sum_{i=1}^{n}\sum_{j=1}^{n}\left(\frac{1}{n^2}k_Z(\boldsymbol{z}^{(i)}, \boldsymbol{z}^{(j)})k_Y(\boldsymbol{y}^{(i)}, \boldsymbol{y}^{(j)})\right)^2 \\
&= \frac{1}{n^4}\sum_{i=1}^{n}\sum_{j=1}^{n}k_Z(\boldsymbol{z}^{(i)}, \boldsymbol{z}^{(j)})^2 k_Y(\boldsymbol{y}^{(i)}, \boldsymbol{y}^{(j)})^2,
\end{aligned} \tag{33}$$

Now, we want to compute the gradient of $\mathcal{L}_{\text{Cond-IRKE}}(Z; Y)$ with respect to $\boldsymbol{z}^{(n)}$ according to Eq. (33):

$$\nabla_{\boldsymbol{z}^{(n)}}\mathcal{L}_{\text{Cond-IRKE}}(Z; Y) = \frac{1}{n^4}\nabla_{\boldsymbol{z}^{(n)}}\left(\sum_{i=1}^{n}\sum_{j=1}^{n}k_Z(\boldsymbol{z}^{(i)}, \boldsymbol{z}^{(j)})^2 k_Y(\boldsymbol{y}^{(i)}, \boldsymbol{y}^{(j)})^2\right), \tag{34}$$

Let $S_C = \sum_{i=1}^{n}\sum_{j=1}^{n}k_Z(\boldsymbol{z}^{(i)}, \boldsymbol{z}^{(j)})^2 k_Y(\boldsymbol{y}^{(i)}, \boldsymbol{y}^{(j)})^2$. We need to find $\nabla_{\boldsymbol{z}^{(n)}}S_C$. The terms in the sum $S_C$ that involve $\boldsymbol{z}^{(n)}$ (through $k_Z$) are those where $i = n$ or $j = n$ (or both), and the terms $k_Y(\boldsymbol{y}^{(i)}, \boldsymbol{y}^{(j)})$ do not depend on $\boldsymbol{z}^{(n)}$. We can split the sum as follows:

$$\begin{aligned}
S_C = &\sum_{i=1}^{n-1}\sum_{j=1}^{n-1}k_Z(\boldsymbol{z}^{(i)}, \boldsymbol{z}^{(j)})^2 k_Y(\boldsymbol{y}^{(i)}, \boldsymbol{y}^{(j)})^2 \quad \text{(terms not involving } \boldsymbol{z}^{(n)}) \\
&+ \sum_{i=1}^{n-1}k_Z(\boldsymbol{z}^{(i)}, \boldsymbol{z}^{(n)})^2 k_Y(\boldsymbol{y}^{(i)}, \boldsymbol{y}^{(n)})^2 \quad (j = n, i \neq n) \\
&+ \sum_{j=1}^{n-1}k_Z(\boldsymbol{z}^{(n)}, \boldsymbol{z}^{(j)})^2 k_Y(\boldsymbol{y}^{(n)}, \boldsymbol{y}^{(j)})^2 \quad (i = n, j \neq n) \\
&+ k_Z(\boldsymbol{z}^{(n)}, \boldsymbol{z}^{(n)})^2 k_Y(\boldsymbol{y}^{(n)}, \boldsymbol{y}^{(n)})^2, \quad (i = n, j = n)
\end{aligned} \tag{35}$$

Using the symmetry of the kernels, $k_Z(\boldsymbol{z}^{(n)}, \boldsymbol{z}^{(j)}) = k_Z(\boldsymbol{z}^{(j)}, \boldsymbol{z}^{(n)})$ and $k_Y(\boldsymbol{y}^{(n)}, \boldsymbol{y}^{(j)}) = k_Y(\boldsymbol{y}^{(j)}, \boldsymbol{y}^{(n)})$, the second and third sums are identical. Then we have:

$$\begin{aligned}
S_C = &\sum_{i=1}^{n-1}\sum_{j=1}^{n-1}k_Z(\boldsymbol{z}^{(i)}, \boldsymbol{z}^{(j)})^2 k_Y(\boldsymbol{y}^{(i)}, \boldsymbol{y}^{(j)})^2 \\
&+ 2\sum_{i=1}^{n-1}k_Z(\boldsymbol{z}^{(i)}, \boldsymbol{z}^{(n)})^2 k_Y(\boldsymbol{y}^{(i)}, \boldsymbol{y}^{(n)})^2 + k_Z(\boldsymbol{z}^{(n)}, \boldsymbol{z}^{(n)})^2 k_Y(\boldsymbol{y}^{(n)}, \boldsymbol{y}^{(n)})^2,
\end{aligned} \tag{36}$$

Now, we compute the gradient of $S_C$ with respect to $\boldsymbol{z}^{(n)}$:

$$\begin{aligned}
\nabla_{\boldsymbol{z}^{(n)}}S_C = &\nabla_{\boldsymbol{z}^{(n)}}\left(\sum_{i=1}^{n-1}\sum_{j=1}^{n-1}k_Z(\boldsymbol{z}^{(i)}, \boldsymbol{z}^{(j)})^2 k_Y(\boldsymbol{y}^{(i)}, \boldsymbol{y}^{(j)})^2\right) \\
&+ \nabla_{\boldsymbol{z}^{(n)}}\left(2\sum_{i=1}^{n-1}k_Z(\boldsymbol{z}^{(i)}, \boldsymbol{z}^{(n)})^2 k_Y(\boldsymbol{y}^{(i)}, \boldsymbol{y}^{(n)})^2\right) \\
&+ \nabla_{\boldsymbol{z}^{(n)}}\left(k_Z(\boldsymbol{z}^{(n)}, \boldsymbol{z}^{(n)})^2 k_Y(\boldsymbol{y}^{(n)}, \boldsymbol{y}^{(n)})^2\right),
\end{aligned} \tag{37}$$

The first term is zero because it does not depend on $\boldsymbol{z}^{(n)}$. For the second term:

$$
\begin{aligned}
\nabla_{\boldsymbol{z}^{(n)}}\left(2\sum_{i=1}^{n-1}k_Z(\boldsymbol{z}^{(i)},\boldsymbol{z}^{(n)})^2 k_Y(\boldsymbol{y}^{(i)},\boldsymbol{y}^{(n)})^2\right) &= 2\sum_{i=1}^{n-1}k_Y(\boldsymbol{y}^{(i)},\boldsymbol{y}^{(n)})^2\nabla_{\boldsymbol{z}^{(n)}}\left(k_Z(\boldsymbol{z}^{(i)},\boldsymbol{z}^{(n)})^2\right)\\
&= 2\sum_{i=1}^{n-1}k_Y(\boldsymbol{y}^{(i)},\boldsymbol{y}^{(n)})^2\left(2k_Z(\boldsymbol{z}^{(i)},\boldsymbol{z}^{(n)})\nabla_{\boldsymbol{z}^{(n)}}k_Z(\boldsymbol{z}^{(i)},\boldsymbol{z}^{(n)})\right)\\
&= 4\sum_{i=1}^{n-1}k_Z(\boldsymbol{z}^{(i)},\boldsymbol{z}^{(n)})k_Y(\boldsymbol{y}^{(i)},\boldsymbol{y}^{(n)})^2\nabla_{\boldsymbol{z}^{(n)}}k_Z(\boldsymbol{z}^{(i)},\boldsymbol{z}^{(n)}),
\end{aligned}
\tag{38}
$$

For the third term, since $k_Z(\boldsymbol{z}^{(n)},\boldsymbol{z}^{(n)})=1$ and $k_Y(\boldsymbol{y}^{(n)},\boldsymbol{y}^{(n)})=1$ (normalized kernels), this term is constant:

$$
\nabla_{\boldsymbol{z}^{(n)}}\left(k_Z(\boldsymbol{z}^{(n)},\boldsymbol{z}^{(n)})^2 k_Y(\boldsymbol{y}^{(n)},\boldsymbol{y}^{(n)})^2\right) = \nabla_{\boldsymbol{z}^{(n)}}(1^2\cdot 1^2) = \nabla_{\boldsymbol{z}^{(n)}}(1) = \boldsymbol{0},
\tag{39}
$$

So we have:

$$
\nabla_{\boldsymbol{z}^{(n)}}S_C = 4\sum_{i=1}^{n-1}k_Z(\boldsymbol{z}^{(i)},\boldsymbol{z}^{(n)})k_Y(\boldsymbol{y}^{(i)},\boldsymbol{y}^{(n)})^2\nabla_{\boldsymbol{z}^{(n)}}k_Z(\boldsymbol{z}^{(i)},\boldsymbol{z}^{(n)}),
\tag{40}
$$

Substituting this back into the gradient of $\mathcal{L}_{\text{Cond-IRKE}}(Z;Y)$ in Eq. (33):

$$
\begin{aligned}
\nabla_{\boldsymbol{z}^{(n)}}\mathcal{L}_{\text{Cond-IRKE}}(Z;Y) &= \frac{1}{n^4}\left(4\sum_{i=1}^{n-1}k_Z(\boldsymbol{z}^{(i)},\boldsymbol{z}^{(n)})k_Y(\boldsymbol{y}^{(i)},\boldsymbol{y}^{(n)})^2\nabla_{\boldsymbol{z}^{(n)}}k_Z(\boldsymbol{z}^{(i)},\boldsymbol{z}^{(n)})\right)\\
&= \frac{4}{n^4}\sum_{i=1}^{n-1}k_Z(\boldsymbol{z}^{(i)},\boldsymbol{z}^{(n)})k_Y(\boldsymbol{y}^{(i)},\boldsymbol{y}^{(n)})^2\nabla_{\boldsymbol{z}^{(n)}}k_Z(\boldsymbol{z}^{(i)},\boldsymbol{z}^{(n)}),
\end{aligned}
\tag{41}
$$

Since $4/n^4$ is a positive constant (for $n\geq 1$), we finally get:

$$
\nabla_{\boldsymbol{z}^{(n)}}\mathcal{L}_{\text{Cond-IRKE}} \propto \sum_{i=1}^{n-1}k_Z(\boldsymbol{z}^{(i)},\boldsymbol{z}^{(n)})k_Y(\boldsymbol{y}^{(i)},\boldsymbol{y}^{(n)})^2\nabla_{\boldsymbol{z}^{(n)}}k_Z(\boldsymbol{z}^{(i)},\boldsymbol{z}^{(n)}).
\tag{42}
$$

This proves the form of the gradient.

Regarding the monotonic relationship: From Eq. (9), we have:

$$
\mathcal{L}_{\text{Cond-RKE}}(Z;Y) = \frac{\left\|\widetilde{K}_Y\right\|_F^2}{\left\|\frac{K_Y\odot K_Z}{\text{Tr}(K_Y\odot K_Z)}\right\|_F^2},
\tag{43}
$$

We have shown that $\text{Tr}(K_Y\odot K_Z)=n$, since $(K_Y\odot K_Z)_{ii}=k_Y(\boldsymbol{y}^{(i)},\boldsymbol{y}^{(i)})k_Z(\boldsymbol{z}^{(i)},\boldsymbol{z}^{(i)})=1\cdot 1=1$. Also, $\|\widetilde{K}_Y\|_F^2 = \|K_Y/n\|_F^2 = \|K_Y\|_F^2/n^2$. Substituting these into the formulation for $\mathcal{L}_{\text{Cond-RKE}}$ in Eq. (43):

$$
\mathcal{L}_{\text{Cond-RKE}}(Z;Y) = \frac{\frac{1}{n^2}\|K_Y\|_F^2}{\left\|\frac{K_Y\odot K_Z}{n}\right\|_F^2} = \frac{\frac{1}{n^2}\|K_Y\|_F^2}{\frac{1}{n^2}\|K_Y\odot K_Z\|_F^2} = \frac{\|K_Y\|_F^2}{\|K_Y\odot K_Z\|_F^2},
\tag{44}
$$

From the definition of $\mathcal{L}_{\text{Cond-IRKE}}(Z;Y)$ in Eq. (31), we have:

$$
\mathcal{L}_{\text{Cond-IRKE}}(Z;Y) = \|\widetilde{K}_Z\odot\widetilde{K}_Y\|_F^2 = \left\|\frac{K_Z}{n}\odot\frac{K_Y}{n}\right\|_F^2 = \frac{1}{n^4}\|K_Z\odot K_Y\|_F^2,
\tag{45}
$$

Let $C_Y = \|K_Y\|_F^2$. This term depends only on the conditions $Y$ and is constant with respect to $\boldsymbol{z}^{(n)}$. $C_Y$ is always positive since the kernel function $K_Y$ is positive semi-definite and not all kernel values can be 0. From the formulation for $\mathcal{L}_{\text{Cond-IRKE}}$ in Eq. (31), we have $\|K_Z\odot K_Y\|_F^2 = n^4\mathcal{L}_{\text{Cond-IRKE}}(Z;Y)$. Substituting this into the formulation for $\mathcal{L}_{\text{Cond-RKE}}$:

$$
\mathcal{L}_{\text{Cond-RKE}}(Z;Y) = \frac{C_Y}{n^4\mathcal{L}_{\text{Cond-IRKE}}(Z;Y)}.
\tag{46}
$$

Since $C_Y/n^4$ is a positive constant, then $\mathcal{L}_{\text{Cond-RKE}}(Z;Y)$ is a strictly monotonically decreasing function of $\mathcal{L}_{\text{Cond-IRKE}}(Z;Y)$. Thus, they change monotonically with respect to each other. This completes the proof of Proposition 2. $\qquad\square$

### B.3 Interpretation of SPARKE

Our method, SPARKE, can be theoretically interpreted as an interacting particle system where sequential samples $\mathbf{z}^{(1)}, \mathbf{z}^{(2)}, ..., \mathbf{z}^{(n)}$ evolve along a gradient flow to minimize an energy potential, thus improving prompt-aware diversity. This process is described by the following sequential stochastic differential equation:

$$\mathrm{d}\mathbf{z}^{(n)} = \left[ -\mathbf{f}(\mathbf{z}^{(n)}, t') + \frac{g^2(t')}{2} \left( \nabla_{\mathbf{z}^{(n)}} \log p_{t'}(\mathbf{z}^{(n)}) - \eta \nabla_{\mathbf{z}^{(n)}} L_{\text{Cond-IRKE}}(\mathbf{z}_1, \dots, \mathbf{z}_n) \right) \right] \mathrm{d}t + g(t')\mathrm{d}\mathbf{w},$$
(47)

The core insight of SPARKE is to employ the Conditional Rényi Kernel Entropy (Conditional RKE) as a differentiable objective for promoting diversity. Following [16], the order-2 RKE score serves as a differentiable objective for counting the modes of a particle distribution. Furthermore, the Conditional RKE score in [15] quantifies the internal diversity conditioning on the prompt categories. SPARKE defines a gradient flow that optimizes the Conditional RKE score, and sequentially drives samples to cover distinct modes, while the conditional mechanism ensures the prompt-aware diversity guidance potential.

## C   Implementation Details and Hyperparameters

In the kernel-based guidance experiments of SPARKE and the baselines with kernel entropy diversity scores, we considered a Gaussian kernel, which consistently led to higher output scores in comparison to the other standard cosine similarity kernel (see Section D for ablation studies). We used the same Gaussian kernel bandwidth $\sigma$ in the RKE and Vendi experiments, and the bandwidth parameter choice matches the selected value in [15, 13]. The numerical experiments were conducted on 4×NVIDIA GeForce RTX 4090 GPUs, each of which has 22.5 GB of memory.

### C.1   Experimental Configuration for Table 1

We evaluated the methods listed in Table 1 in the following setting. We used Stable Diffusion 2.1 with a resolution of 1024×1024, a fixed classifier-free guidance scale of $W_{\text{CFG}} = 7.5$, and 50 inference steps using the DPM solver. We used the first 10,000 prompts of the MS-COCO 2014 validation set and fixed the generation seed to be able to compare the effect of the methods. For the methods, we used the following configuration to generate the results reported in Table 1.

The hyperparameter tuning was performed by performing cross-validation on the in-batch similarity score, selecting the hyperparameter values that optimized this alignment-based metric. Note that the in-batch similarity score accounts for both text-image consistency and inter-sample diversity as discussed in [56]. Due to GPU memory requirements (as mentioned in Section B.2 in [56]), we were unable to evaluate the Particle Guide baseline [56] on SD v2.1. Following the provided implementation of this baseline for SD v1.5, we conducted this baseline's experiments only on SD v1.5.

**Stable Diffusion.** We used the standard CFG guidance and DPM solver with no additional diversity-related guidance.

**CADS.** Following the discussion in Table 13 of [14], we set the threshold parameters as $\tau_1 = 0.6$ and $\tau_2 = 0.9$, $\psi = 1$, and used a noise scale of 0.25.

**c-VSG.** We note that the reference [10] considered GeoDE [66] and DollarStreet [73] datasets, in which multiple samples exist per input prompt. On the other hand, in our experiments, we considered the standard MSCOCO prompt set where for each prompt corresponds we access a single image, making the contextualized Vendi guidance baseline in [10] not directly applicable. Therefore, we simulated the non-contextualized version of VSG. For selecting the Vendui score guidance scale, we performed validation over the set $\{0, 0.04, 0.05, 0.06, 0.07\}$, following the procedure in [10]. A guidance frequency of 5 was used, consistent with the original implementation. To maintain stable gradient computation for the Vendi score, we implemented a sliding window of 150 most recently generated samples, as gradient calculations became numerically unstable for some steps beyond this threshold.

**latent RKE Guidance.** We used a Gaussian kernel with bandwidth $\sigma_{img} = 0.8$ and used $\eta = 0.03$ as the weight of RKE guidance. To balance the effects of the diversity guidance in sample generation,

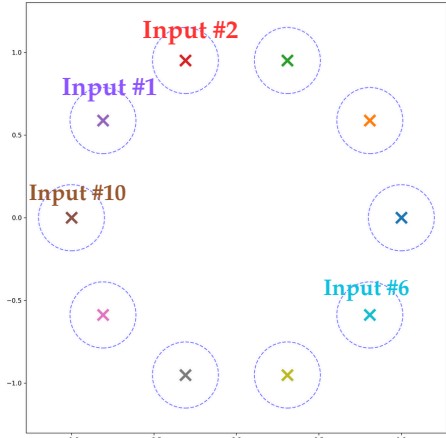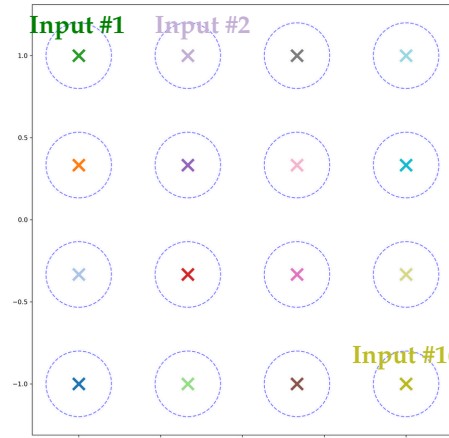

Figure 7: Illustration of input-based Gaussian output used for the conditional diffusion model in Figure 4. As displayed, the output 2D Gaussian vector is generated centered around the mean vector specified by the input prompt number.

the RKE guidance update was applied every 10 reverse-diffusion steps in the diffusion process, which is similar to the implementation of Vendi score guidance in [10]. Unlike VSG, which requires a sample window limit, RKE guidance operates without window size constraints, leveraging the complete history of generated samples for gradient computation.

**SPARKE (latent Conditional RKE Guidance).** We considered the same Gaussian kernel for the image generation with bandwidth $\sigma_{img} = 0.8$ and used bandwidth parameter $\sigma_{text} = 0.3$ for the text kernel. The guidance hyperparameter was set to $\eta = 0.03$, as in RKE guidance. Similar to the RKE and Vendi guidance, the SPARKE diversity guidance was applied every 10 reverse-diffusion steps. Unlike VSG but similar to RKE, SPARKE uses the complete history of generated samples for guidance.

## C.2 Experiment Settings in the results of Table 4 and Figure 4

We conducted additional experiments using Stable Diffusion v1.5 with a resolution of 512×512. We created a prompt set by performing K-Means clustering on the MSCOCO 2014 validation prompts. Specifically, we clustered the MS-COCO prompt dataset via the spectral clustering in the CLIP embedding space, into 40 groups and randomly drawn 50 prompts from each cluster, resulting in a total of 2,000 prompts. To evaluate the performance of our method with the baselines with different seeds, we generated five samples per prompt using seeds 0, 1, 2, 3, and 4, yielding a total of 10,000 images for each method. The complete list of prompts is included in the supplementary materials.

For all the tested methods, we considered the settings described in Section C.1, including a classifier-free guidance scale of $W_{\text{CFG}} = 7.5$, 50 reverse-diffusion steps, and the DPM solver. For the Particle Guide baseline, we used the following settings:

**Particle Guide.** We applied the original implementation provided by the repository of [56]. Since the method operates on repeated generations of the same prompt, we generated five samples per prompt using seeds 0 through 4. The coefficient parameter was set to 30, following the original implementation.

**Gaussian Mixtures.** We used conditional diffusion models in Figure 4. In Figure 7, we illustrate how the centers of clusters were used as inputs to the diffusion model to simulate conditional generation.

## D    Additional Numerical Results

In this section, we provide additional numerical results for the SPARKE guidance method.

**Ablation studies on the kernel function choice in SPARKE.** In Figure 9, we compared the outputs of applying SPARKE with the cosine similarity kernel and the Gaussian kernel in the latent space of

**Stable Diffusion XL**

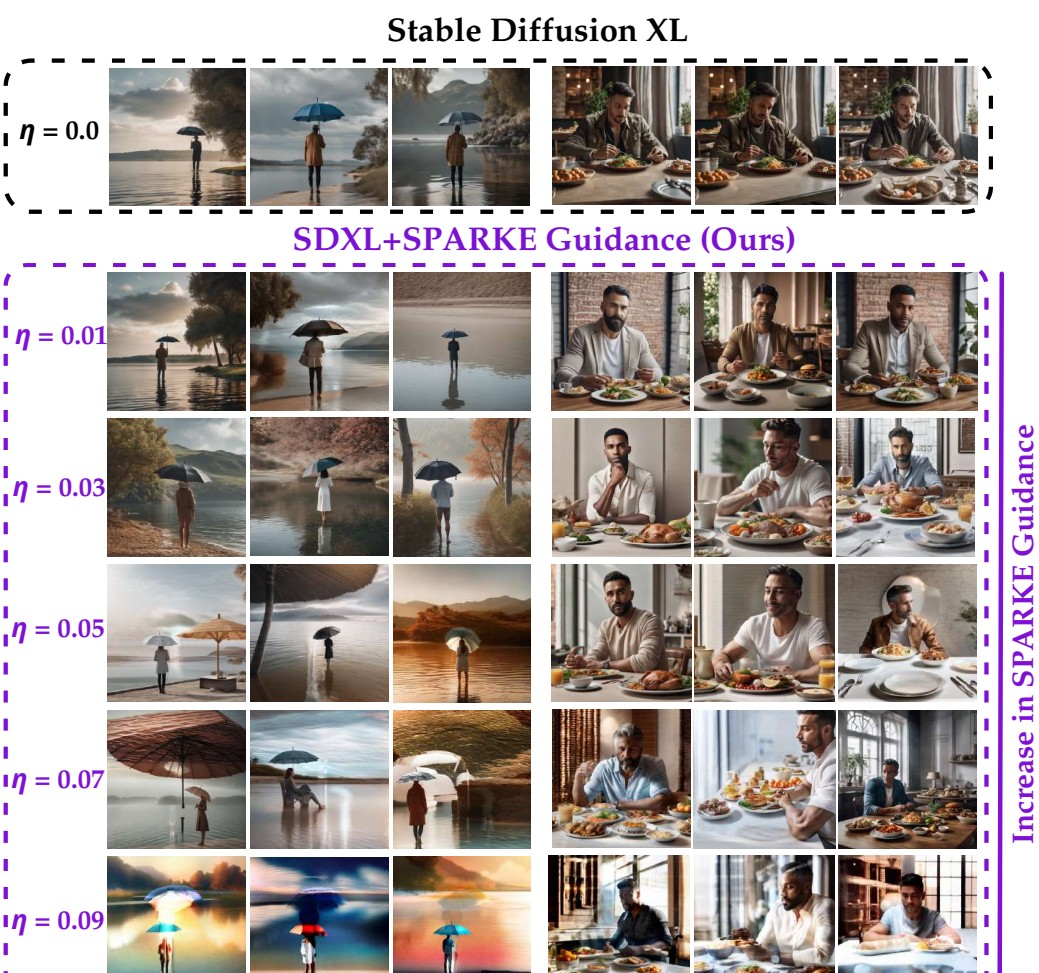

**Figure 8:** Qualitative results for various diversity guidance scales from $\eta = 0$ (no guidance) to $\eta = 0.09$. The results are generated using Stable Diffusion XL with the same Gaussian kernel.

SDXL. As shown in the figure, the Gaussian kernel seems able to exhibit more diverse features, and therefore, the diversity of its images looks higher than the outputs of the cosine similarity kernel.

**Hyperparameter selection for SPARKE.** In Figure 8, we analyzed the impact of different diversity guidance scales $\eta$, ranging from 0 to 0.09. This range was explored to investigate the trade-off between generation quality and diversity. We observed that as $\eta$ gradually increased, there was a corresponding increase in image diversity and a relative decrease in image quality. As a result, we chose $\eta = 0.03$ in our experiments to show the balanced trade-off.

**Effect of Classifier-Free Guidance scale on SPARKE guidance.** We study the Classifier-free guidance's impact on SPARKE for the Stable Diffusion 1.5 model, demonstrating how our method alleviates the known quality-diversity trade-off [52]. We changed the CFG scale from 2 to 8 while using the same setting of SD 1.5 as mentioned in Section C.2. We evaluated SPARKE's effect on the quality-diversity trade-off using Precision/Recall metrics in Figure 12 and Density/Coverage metrics in Figure 11. Our results show that SPARKE guidance maintains diversity in a high CFG scale while preserving acceptable generation quality. Additionally, the Vendi score and Kernel Distance (KD) evaluations in Figure 10 show the same trend. Qualitative comparisons of generated samples at CFG

Gaussian  Kernel        Cosine  Kernel

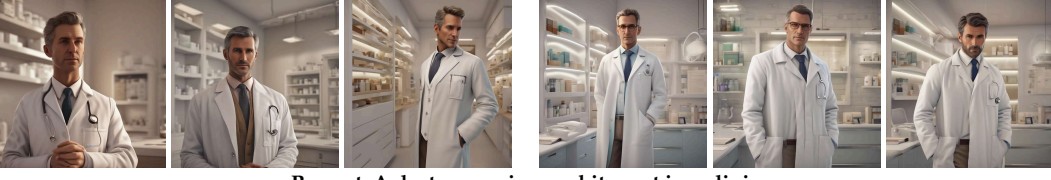

**Prompt: A dog standing lying in front of a crackling fireplace in a chalet in Europe.**

**Prompt: A man casually drinking a hot cup of coffee.**

**Prompt: A cat resting quietly on a wooden chair.**

**Prompt: A dog standing beside a cherry blossom tree in full bloom in East Asia.**

**Prompt: A doctor wearing a white coat in a clinic.**

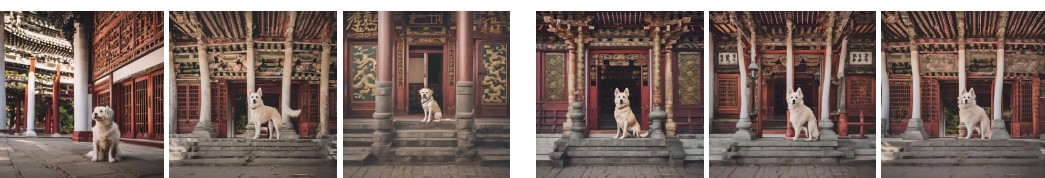

**Prompt: A dog standing guarding the entrance of a historic palace in East Asia.**

Figure 9: Qualitative comparison between SPARKE numerical results applying Gaussian (RBF) kernel and Cosine similarity kernel on Stable Diffusion XL.

scales $w = 4, 6, 8$ are presented in Figure 13, demonstrating that SPARKE produces diverse outputs without compromising quality.

**Additional quantitative comparison with the baseline diversity-guided diffusion-based sample generations.** Similar to Table 1, we compare different guidance methods on Stable Diffusion 1.5 [60] in Table 4.

**Latent space vs. ambient space.** We compared latent-space guidance (as in SPARKE) with the CLIP-embedded ambient-space guidance in c-VSG [10] using three categories from the GeoDE dataset [66] and modified it with GPT-4o [74]. We use a similar prompt template as mentioned in Table 7 in [10] to add more details to the prompt. We provide additional samples in Figure 14. We

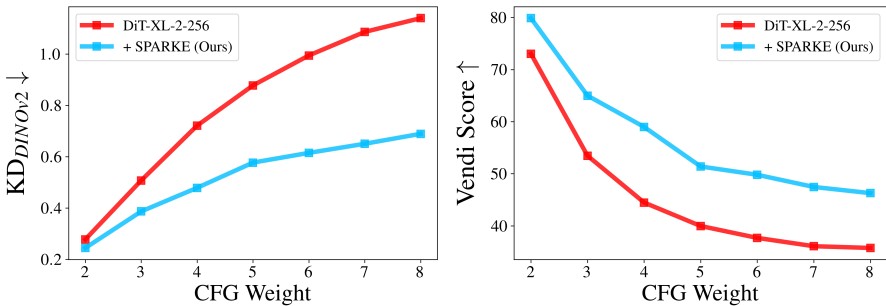

Figure 10: Comparison of image generation diversity and quality in the DiT-XL-2-256 model, analyzing the impact of SPARKE guidance through Vendi Score and Kernel Distance.

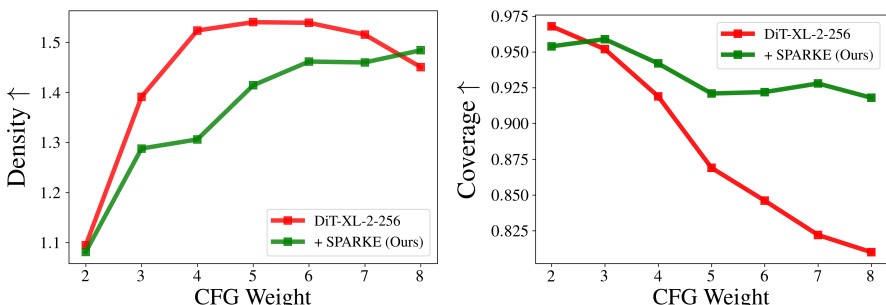

Figure 11: Comparison of image generation quality and diversity in the DiT-XL-2-256 model, analyzing the impact of SPARKE guidance through density and coverage metrics.

also provide quantitative results in Table 5 shows that although CLIP-based Vendi scores were slightly higher for CLIP-embedded ambient guidance, which could be due to optimizing Vendi with the CLIP features, the DINOv2-based Vendi scores and in-batch similarity scores indicate higher diversity for the latent Vendi entropy guidance that we proposed for LDMs. Furthermore, latent guidance can be performed with a considerably lower GPU memory (in our implementation $\approx$20GB for the latent case vs. $approx$35 GB for the ambient case), resulting in higher computational efficiency.

We use the following template:

Prompt to GPT-4o: "You are an expert prompt optimizer for text-to-image models. Text-to-image models take a text prompt as input and generate images depicting the prompt as output. You translate prompts written by humans into better prompts for the text-to-image models. Your answers should be concise and effective. Your task is to optimize this prompt template written by a human: "object in region". This prompt template is used to generate many images of objects such as dogs, chairs, and cars in regions such as Africa, Europe, and East Asia. Generate one sentence of the initial prompt templates that contains the keywords "object" and "region" but increases the diversity of the objects depicted in the image."

**Additional qualitative comparisons of vanilla, baseline diversity guided, and SPARKE prompt-aware diversity guided diffusion models.** We compared the SPARKE prompt-aware diversity guidance method with state-of-the-art conditional latent diffusion models, including Stable Diffusion 2.1, Stable Diffusion XL [5] and PixArt-$\Sigma$ [61].

Additional qualitative comparisons between prompt-aware (conditional RKE score) and prompt-unaware (RKE score) diversity guidance methods are presented on SD 2.1 (Figure 15). Building on these observations, our results demonstrate that SPARKE improves prompt-aware diversity more effectively than prompt-unaware RKE guidance, as quantified by In-batch Similarity [56].

Furthermore, we showed additional results applied to PixArt-$\Sigma$ (Figure 16) and Stable Diffusion XL (Figure 17).The qualitative comparison of SPARKE and without guidance shows SPARKE guidance significantly improves prompt-aware diversity relative to the baseline, highlighting its efficacy across varied latent diffusion model architectures.

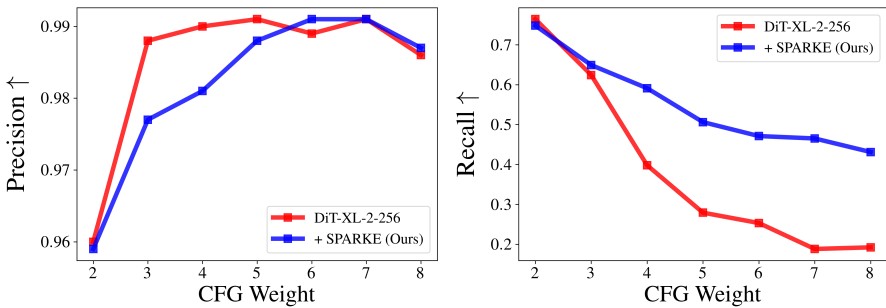

Figure 12: Comparison of image generation quality and diversity in the DiT-XL-2-256 model, analyzing the impact of SPARKE guidance through precision and recall metrics.

Table 4: Quantitative comparison of guidance methods on Stable Diffusion 1.5 (Table 2 in the main text, extended to SD 1.5).

| Method | CLIPScore $\uparrow$ | KD$\times 10^2 \downarrow$ | AuthPct $\uparrow$ | Cond-Vendi $\uparrow$ | Vendi $\uparrow$ | In-batch Sim.$\times 10^2 \downarrow$ |
|---|---|---|---|---|---|---|
| SD v1.5 [60] | 30.15 | 1.045 | 73.86 | 25.41 | 350.28 | 81.25 |
| c-VSG [10] | 27.80 | 1.078 | 74.92 | 26.78 | 357.39 | 79.71 |
| CADS [14] | 28.94 | 1.049 | 75.44 | 27.60 | 360.99 | 78.23 |
| Particle Guide [56] | 29.50 | 1.056 | 72.58 | 27.24 | 345.50 | 79.50 |
| **latent RKE (Ours)** | 29.23 | 1.081 | 78.02 | 28.75 | 369.50 | 78.30 |
| **SPARKE (Ours)** | 29.31 | 1.071 | 80.92 | 31.44 | 386.42 | 76.67 |

Table 5: Comparison of diversity metrics between (CLIP-embedded) ambient and latent guidance.

| Guidance Method | IS $\uparrow$ | Vendi Score$_{\text{CLIP}}$ $\uparrow$ | Vendi Score$_{\text{DINOv2}}$ $\uparrow$ | In-batch Sim.$\downarrow$ |
|---|---|---|---|---|
| (CLIP-embedded) Ambient VSG [10] | 5.45 | 5.34 | 13.33 | 0.28 |
| Latent VSG | 7.24 | 4.42 | 25.22 | 0.15 |

**DiT-XL-2 Model**   **DiT-XL-2 + SPARKE (Ours)**

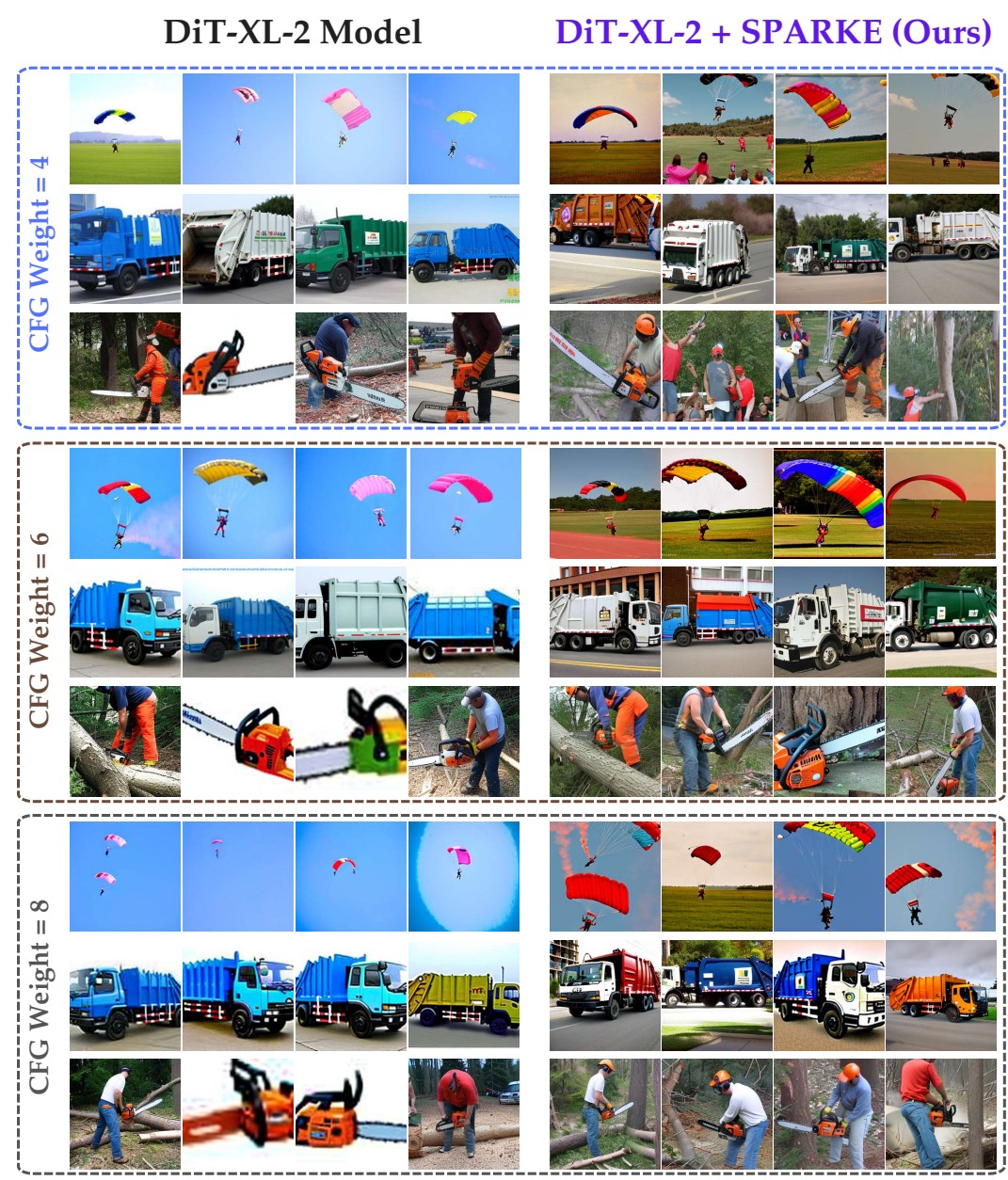

Figure 13: Qualitative comparison of DiT-XL-2 with different classifier free guidance scales: (Left side) standard DiT-XL-2 (Right side) DiT-XL-2 with SPARKE prompt-aware diversity guidance.

**latent Vendi Score Guidance**        **CLIP-(Ambient) Vendi Score Guidance (c-VSG)**

**Prompt: A dog standing beside a village hut in Kenya in Africa.**

**Prompt: A dog standing chasing birds along a cobblestone street in Europe.**

**Prompt: A dog standing beside a grand Gothic cathedral in Europe.**

**Prompt: A dog standing beside a cherry blossom tree in full bloom in East Asia.**

**Prompt: A dog standing guarding a livestock enclosure in Africa.**

Figure 14: Comparison of latent entropy guidance vs. CLIP-(ambient) entropy guidance on SD-XL. Additional Samples for Figure 2.

# Stable Diffusion 2.1

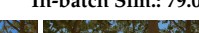

**SPARKE: Prompt-Aware Diversity Guidance**
**(Conditional RKE score)**
In-batch Sim. : 75.68

**Prompt-Unaware Diversity Guidance**
**(RKE score)**
In-batch Sim.: 79.01

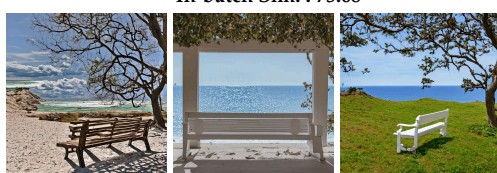 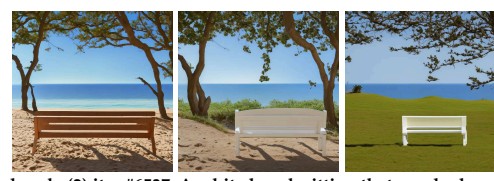

Prompts: (1) iter #6219: This is an image of a bench overlooking a beach, (2) iter #6537: A white bench sitting that overlooks an ocean view, (3) iter #9353.A white wooden bench sitting on top of a green hill next to the ocean.

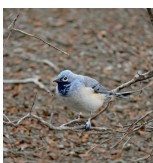 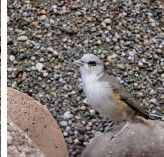 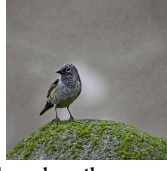 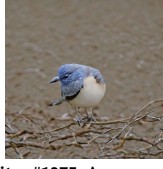 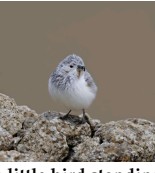 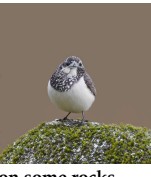

Prompts: (1) iter #1300: A bird on a branch on the ground, (2) iter #1375: A very cute little bird standing on some rocks, (3) iter #3234: A small bird standing on top of a rock near grass.

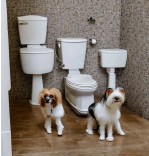 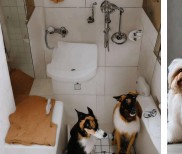 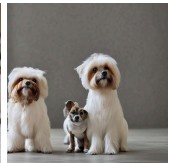 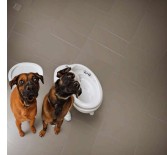 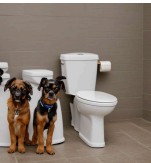 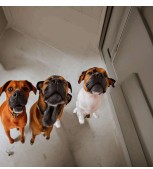

Prompts: (1) iter #242: Two dogs are looking up while they stand near the toilet in the bathroom, (2) iter #3475: Two small dogs standing in a restroom next to a toilet, (3) iter #4875: Small dogs standing in a restroom next to a toilet.

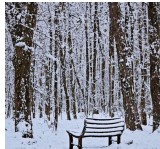 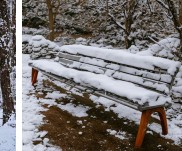 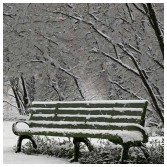 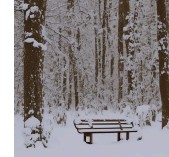 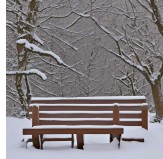 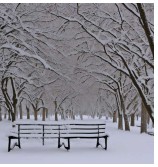

Prompts: (1) iter #6365: A wooden bench with snow in a forest, (2) iter #7348: a wood bench is outside covered in snow, (3) iter #9237: A winter scene of a park bench covered in snow among the trees

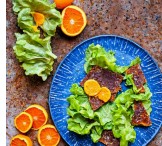 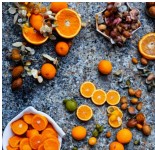 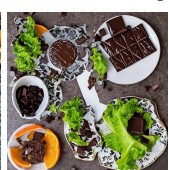 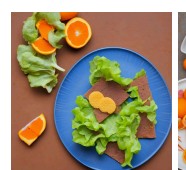 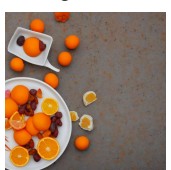 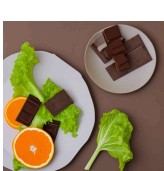

Prompts: (1) iter #7927: a blue plate with an orange a cracker some lettuce and a twist bar, (2) iter #8354: A dish contains an orange and snacks., (3) iter #9287: An orange, chocolate, cracker, and piece of lettuce on a plate

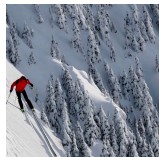 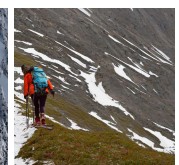 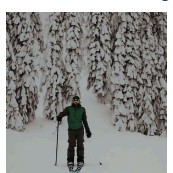 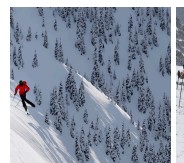 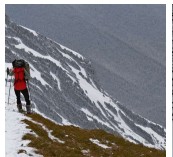 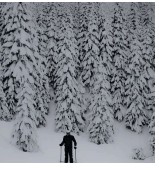

Prompts: (1) iter #8034: A person wearing winter gears skis down a mountain, (2) iter #8527: A woman hiking across a mountain with skis., (3) iter #9275: A man on ski's that is standing in the snow.

Figure 15: Comparison of SPARKE prompt-aware diversity guidance via conditional-RKE score vs. diversity-unaware diversity guidance using the RKE score on SD 2.1 text-to-image generation. Additional Samples for Figure 5.

**PixArt-Σ**                    **PixArt-Σ + SPARKE Guidance**

Prompts: 1. A photo of a man posing for a portrait, 2. A portrait of a young male facing forward, 3. An upper body portrait of a man.

Prompts: 1. A relaxed cat sitting still on a chair, 2. A fluffy cat sitting calmly on a seat, 3. A peaceful cat on top of a simple chair.

Prompts: 1. A bird sitting calmly among the branches, 2. A bird resting peacefully on a twig, 3. A tree branch holding a bird.

Prompts: 1. An old man walking down a street, 2. An elderly man strolling through a street, 3. An old gentleman taking a walk on the road.

Prompts: 1. A small kid coloring on a table, 2. A child sitting and drawing on a piece of paper, 3. A kid making a drawing on a paper.

Prompts: 1. A man sitting quietly and reading a book, 2. A man reading book on a bench, 3. A man reading a book in a relaxed setting

Prompts: 1. A young woman holding an apple, 2. A female holding an apple and smiling, 3. A woman showing apple with both hands.

Figure 16: Qualitative Comparison of samples generated by standard PixArt-Σ vs. SPARKE diversity guided PixArt-Σ.

**SDXL**                    **SDXL + SPARKE Guidance**

Prompts: 1. A photo of a man posing for a portrait, 2. A portrait of a young male facing forward, 3. An upper body portrait of a man.

Prompts: 1. A relaxed cat sitting still on a chair, 2. A fluffy cat sitting calmly on a seat, 3. A peaceful cat on top of a simple chair.

Prompts: 1. A bird sitting calmly among the branches, 2. A bird resting peacefully on a twig, 3. A tree branch holding a bird.

Prompts: 1. An old man walking down a street, 2. An elderly man strolling through a street, 3. An old gentleman taking a walk on the road.

Prompts: 1. A small kid coloring on a table, 2. A child sitting and drawing on a piece of paper, 3. A kid making a drawing on a paper.

Prompts: 1. A man sitting quietly and reading a book, 2. A man reading book on a bench, 3. A man reading a book in a relaxed setting

Prompts: 1. A young woman holding an apple, 2. A female holding an apple and smiling, 3. A woman showing apple with both hands.

Figure 17: Qualitative Comparison of samples generated by standard Stable Diffusion XL vs. SPARKE diversity guided Stable Diffusion XL.

