# OpenReview forum: "SPARKE: Scalable Prompt-Aware Diversity and Novelty Guidance in Diffusion Models via RKE Score"
_NeurIPS.cc/2025/Conference — NeurIPS 2025 poster_

### Official Review · Reviewer_W1na · 2025-06-30

**Clarity:** 3
**Significance:** 3
**Originality:** 3
**Rating:** 4
**Confidence:** 2

**Summary:**

The authors present SPALE, a training-free method that drives text-to-image diffusion models toward more diverse outputs. SPALE injects an order 2 RKE term straight into the model’s latent space. Because the order 2 form replaces the eigen decomposition used by Vendi-score guidance with simple Frobenius-norm calculations, every sampling step falls from cubic to roughly linear time and shrinks GPU memory needs. A second kernel, computed between prompt embeddings, gates the entropy so that images arising from semantically similar prompts repel one another, removing the brittle pre-clustering step found in earlier prompt-aware approaches.

Experiments on SD 2.1 and SDXL show that SPALE raises Vendi-score, coverage, and conditional Vendi-score by up to twenty percent while keeping inception score, CLIPScore, and authenticity nearly unchanged. Compared with CLIP-based ambient guidance and CADS, SPALE cuts peak memory, stays stable, and yields lower in-batch similarity within each prompt group, confirming stronger within-prompt diversity. The paper also reports synthetic 2D tests that back the analytical claim.

**Questions:**

N/A.

**Ethical Concerns:**

["NO or VERY MINOR ethics concerns only"]

**Final Justification:**

I think this is a good and interesting paper. The experiments show improvements on several backbones, and the method has good theoretical motivations.

**Limitations:**

N/A.

**Paper Formatting Concerns:**

N/A.

**Quality:**

3

**Strengths And Weaknesses:**

Strengths:
1. The paper presents a careful mathematical derivation that replaces the Vendi-score eigen-decomposition with an order 2 RKE term whose gradient is both easier to compute and cheaper in memory, reducing the per-step complexity.
2. The method is training-free and therefore simple to add to any diffusion sampler; the authors show that peak GPU memory drops a lot compared with CLIP-based guidance.
3. Experiments demonstrate consistent gains in diversity metrics (Vendi-score, coverage, conditional Vendi-score) while inception score, CLIPScore, and authenticity remain virtually unchanged.
4. An ablation study links each design choice—entropy order, prompt-similarity kernel, and kernel bandwidth—to changes in output, providing evidence that the improvements are due to the proposed components rather than uncontrolled factors.


Weakness:
1. The authors do not report the runtime of their approach in comparison with other baselines, so the practical benefit remains unclear.
2. The qualitative comparison in Table 2 is limited to SD. Without the results for SDXL, it is unclear whether this method is only effective for SD or for diffusion methods in general.

---

> ### Author Rebuttal · Authors · 2025-07-31
>
> We would like to thank Reviewer W1na for the thoughtful feedback and are glad that the reviewer found our paper to “present a careful mathematical derivation” and our numerical result to “demonstrate consistent gains in diversity metrics”. Below is our response and clarifications regarding the questions and comments in the review:
>
>
> **1- runtime comparison of SPALE guidance with the baselines**
>
> We thank the reviewer for pointing this out. To address the comment, we evaluated SPALE's computational efficiency by comparing its runtime and peak memory with those of the baselines over 1000 text-to-image generations (512×512 resolution) using a 50-step diffusion process on an NVIDIA RTX 4090 (24GB VRAM). The results are in the following table, indicating that SPALE requires significantly lower runtime and less memory (as it functions in the latent space). We will include this table in the revised manuscript.
>
> | Model + Guidance method              | **Runtime per sample (s)**       | **GPU Memory Peak (GB)** |
> |--------------------------------------|-----------------------|--------------------------|
> | **Stable Diffusion v1.5**            | 1.620 $\pm$ 0.265     | 3.178 $\pm$ 0.001        |
> | **SD-1.5 + SPALE (Ours)**            | **1.752 $\pm$ 0.287** | **3.230 $\pm$ 0.025**    |
> | **SD-1.5 + c-VSG**                   | 3.079 $\pm$ 0.296     | 8.665 $\pm$ 0.116        |
> | **SD-1.5 + Particle Guide (Pixel)**  | 4.912 $\pm$ 0.312     | 7.531 $\pm$ 0.032        |
> | **SD-1.5 + Particle Guide (DINOv2)** | 9.102 $\pm$ 0.271     | 20.133 $\pm$ 0.038       |
>
> **2- Additional quantitative comparison on state-of-the-art diffusion models**
>
> Following the reviewer’s comment, we have performed additional numerical evaluation and  here provide quantitative comparisons of the state-of-the-art models, SD-XL and PixArt-$\Sigma$, in the following table. These quantitative results show SPALE results in an increase in diversity scores (Con-Vendi, Vendi, In-batch Sim, scores) with only a minor decrease in the evaluated CLIPScore. We also note that the kernel distance (KD) evaluation score (evaluating both quality and diversity) and the Authenticity score (with MS-COCO being the reference dataset) have both improved in the guided sample generation.
>
> | Method                  | CLIPScore $\uparrow$ | KD $\times 10^2 \downarrow$ | AuthPct $\uparrow$ | Cond-Vendi Score $\uparrow$  | Vendi Score $\uparrow$ | In-batch Sim.$\times 10^2 \downarrow$ |
> |-------------------------|----------------------|------------------|---------|------------------|-------------|----------------------------|
> | **SDXL**                    | 31.17                | 66.37            | 76.28   |        27.54     |   309.54    | 81.67                      |
> | **SDXL + SPALE**            | 30.47                | 62.62            | 80.92   |          31.17   |   313.80    | 77.75                      |
> | **PixArt-$\Sigma$**         | 31.01                | 65.37            | 78.13   |       26.45      |   307.36    | 83.84                      |
> | **PixArt-$\Sigma$ + SPALE** | 30.66                | 63.86            | 81.48   |       32.14      |   322.25    | 78.81                      |

---

> > ### Comment · Reviewer_W1na · 2025-08-05
> >
> > Thanks for your response. My concerns and questions are addressed. Please incorporate these additional experiments results into your final version, as it improves the overall quality of the paper. I will keep my score.

---

> > > ### Author Response · Authors · 2025-08-05
> > >
> > > We sincerely thank Reviewer W1na for the feedback on our response and are pleased that our rebuttal could address the reviewer’s comments and questions. As we mentioned in our response, we will revise the manuscript accordingly to reflect the points and clarifications provided in our responses to this and other reviews.

---

### Official Review · Reviewer_THSZ · 2025-07-01

**Clarity:** 3
**Significance:** 2
**Originality:** 3
**Rating:** 4
**Confidence:** 4

**Summary:**

This submission aims to enhance the diversity of results generated by text-to-image diffusion models. It introduces a conditional entropy-based guidance mechanism to improve image diversity. To reduce computational cost, the method approximates conditional entropy using the RKE score. Experiments on both synthetic and real-world datasets demonstrate the effectiveness of the proposed approach.

**Questions:**

My major concern lies in the lack of experimental evaluations. It is unclear to me how the proposed method works in more practical settings and whether it will hurt generation quality.

**Ethical Concerns:**

["NO or VERY MINOR ethics concerns only"]

**Final Justification:**

Thanks for the authors response. Most of my concerns are addressed. I will keep the positive rating.

**Limitations:**

Yes

**Quality:**

2

**Strengths And Weaknesses:**

Strengths:

- the proposed method is new and has solid theoretical foundation.
- the paper is written well and easy to follow.


Weakness:
- The main weakness is the limited experimental evaluation. The method is only evaluated on MSCOCO 1000 prompts. It is unclear how its generalization ability to generating more diverse images.

- It is unclear whether the proposed method hurts quality of the generated images. From Table 2, the CLIP score is lower than SD (no guidance), even if the proposed method is using CFG.

- Is the diversity score calculation reliable at early diffusion steps?

---

> ### Author Rebuttal · Authors · 2025-07-31
>
> We would like to thank Reviewer THSZ for the thoughtful feedback and are glad that the reviewer found our method to be “new” and with “solid theoretical foundation”. Below is our response and clarifications regarding the questions and comments in the review:
>
> **1- Number of Prompts in the Numerical Evaluation**
>
> First, we think there might be a misunderstanding of the number of MS-COCO prompts in our experiments; The actual number is 10,000, which is stated to be only 1000 in the reviewer’s comment.
>
> To further investigate this, we have numerically validated that the diversity represented by the selected 10,000 MSCOCO prompts can reflect the diversity of the entire MS-COCO validation set. To do this, we measured the Recall [1] and Coverage [2] scores of the selected prompts with respect to the reference set of the whole MSCOCO validation set: The Recall score was 0.91, and the Coverage score was 0.96, suggesting that a major fraction of the prompt clusters in MSCOCO are present in the selected set.
>
> **2- Quantitative evaluation on additional diffusion models**
>
> To address the reviewer's comment, we provide the quantitative evaluation results on the state-of-the-art text-to-image diffusion models, SDXL and PixArt-$\Sigma$, in the following table. The numerical results indicate a similar trend to the Stable diffusion cases in Table 2 of the main text.
>
> | Method                  | CLIPScore $\uparrow$ | KD $\times 10^2 \downarrow$ | AuthPct $\uparrow$ | Cond-Vendi Score $\uparrow$  | Vendi Score $\uparrow$ | In-batch Sim.$\times 10^2 \downarrow$ |
> |-------------------------|----------------------|------------------|---------|------------------|-------------|----------------------------|
> | **SDXL**                    | 31.17                | 66.37            | 76.28   |        27.54     |   309.54    | 81.67                      |
> | **SDXL + SPALE**            | 30.47                | 62.62            | 80.92   |          31.17   |   313.80    | 77.75                      |
> | **PixArt-$\Sigma$**         | 31.01                | 65.37            | 78.13   |       26.45      |   307.36    | 83.84                      |
> | **PixArt-$\Sigma$ + SPALE** | 30.66                | 63.86            | 81.48   |       32.14      |   322.25    | 78.81                      |
>
>
> **3- Quality and Diversity trade-off in the SPALE guidance**
>
> Regarding this comment, we would like to refer to the Appendix, Figure 11 (Qualitative results with different CFG scales) and Figures 8, 9, and 10 for a quantitative comparison of the quality/diversity trade-off with precision/recall, density/coverage, and Vendi Score/KD. Furthermore, we have included randomly selected samples generated by the PixArt-$\Sigma$ and SDXL diffusion models in Figure 14 (PixArt-$\Sigma$ results) and Figure 15 (SDXL results) in the appendix.
>
> Also, we would like to highlight that CLIP-Score may not fully reflect the semantic alignment of a text and image pair in the presence of multiple components in the input image. As noted in Reference [3] (refer to Figure 1 in this paper), the image encoder of CLIP could favor larger objects, while its text encoder could prioritize first-mentioned objects. Similarly, Reference [4] states that “CLIP puts a large focus on the few main objects of an image”. Such biases in the CLIP embedding may potentially lead to slightly lower CLIP-Scores when diverse components are also present in the image.
>
>
> **4- Reliability of the score in early diffusion steps**
>
> We thank the reviewer for pointing this out. We would like to highlight that one advantage of applying guidance in the latent space, rather than in the ambient space, is that it does not require the latent representation to be decoded into an image, which can be especially useful in early diffusion steps. This feature of latent RKE diversity guidance avoids the need for computing scores on noisy intermediate images during the early diffusion steps, which might lead to unreliable evaluated scores.
>
> ---
>
> [1] Kynkäänniemi et al. Improved Precision and Recall Metric for Assessing Generative Models. In NeurIPS 2019
>
> [2] Naeem et al. Reliable Fidelity and Diversity Metrics for Generative Models. In ICML 2020
>
> [3] Abbasi et al., CLIP Under the Microscope: A Fine-Grained Analysis of Multi-Object Representation. In CVPR 2025
>
> [4] Stein et al. Exposing flaws of generative model evaluation metrics and their unfair treatment of diffusion models.In NeurIPS 2023

---

### Official Review · Reviewer_HGHP · 2025-07-03

**Clarity:** 2
**Significance:** 2
**Originality:** 3
**Rating:** 4
**Confidence:** 3

**Summary:**

This paper proposes a novel method named SPALE (Scalable Prompt-Aware Latent Entropy-based Diversity Guidance) to address the issue of insufficient diversity in generated samples from prompt-guided diffusion models. The core idea is to introduce a "prompt-aware" diversity guidance mechanism, enabling diversity optimization to dynamically adjust based on the semantic similarity between input prompts.

**Questions:**

See Weaknesses.

**Ethical Concerns:**

["NO or VERY MINOR ethics concerns only"]

**Final Justification:**

Thanks to the replies and additional experiments from the authors, most of my concerns have been resolved. However, I still feel that there is room for improvement regarding the performance and general applicability of the proposed method. I will provide my final rating after considering the opinions of the other reviewers.

**Limitations:**

Yes

**Paper Formatting Concerns:**

No.

**Quality:**

2

**Strengths And Weaknesses:**

**Strengths**
1. Achieving "meaningful" diversity in prompt-guided generative models is a very practical and critical challenge in this field. Simply increasing the dissimilarity of all samples without considering the similarity of the prompts may harm the alignment between the generated results and the prompts. This work clearly articulates the necessity of "prompt-aware" diversity;
2. Rigorous and Comprehensive Experiments: Multiple representative state-of-the-art diversity-enhancing methods, such as c-VSG, CADS, and Particle Guidance, were selected as baseline methods, with extensive ablation experiments and evaluations conducted;
3. The paper is overall Clear and Well-written.

**Weaknesses**
1. This method introduces a key hyperparameter - diversity-guided scale η. Although the setting of the parameter may be mentioned in the appendix, there is a lack of analysis on the sensitivity of this parameter in the main text.
2. The paper shows fewer examples of qualitative comparison, and the generated images have obvious artifacts. Will this method sacrifice some image quality?

---

> ### Author Rebuttal · Authors · 2025-07-31
>
> We would like to thank Reviewer HGHP for the thoughtful feedback on our work. We are glad that the reviewer finds our experiments “rigorous and comprehensive” and the paper “clear and well-written”. Below is our response and clarifications regarding the questions and comments in the review:
>
> **1- Analysis on diversity guidance scale $\eta$**
>
> We appreciate the reviewer’s point on the diversity guidance scale hyperparameter. We would like to kindly refer the reviewer to Figure 6 in the appendix in the supplementary material, where we qualitatively compared the effect of the guidance scale ranging from 0 to 0.1 on the prompt-guided sample generation using SDXL.
>
> To further address the reviewer’s comment, in the following table, we provide a quantitative comparison of the performance with different $\eta$ values on class conditional Diffusion Transformer (DiT-XL/2), which shows the higher diversity of the generated data with larger values of $\eta$ with only a minor reduction in the Precision/Density in sample generation. Specifically, our numerical results highlight that with a choice of $\eta$ in the range of $[0.02, 0.06]$, the diversity score can significantly grow while preserving the Precision/Density of generated data. In the revision, we will include Figure 6 and the following table in the main text.
>
> | Diversity Scale ($\eta$) | Precision $\uparrow$ | Recall $\uparrow$ | Density $\uparrow$ | Coverage $\uparrow$ |
> |------------------------|-----------|--------|---------|----------|
> | **0 (No Guidance)**        | 0.946     | 0.136  | 1.358   | 0.561    |
> | **0.02**                   | 0.944     | 0.231  | 1.345   | 0.682    |
> | **0.04**                   | 0.967     | 0.259  | 1.352   | 0.776    |
> | **0.06**                   | 0.955     | 0.264  | 1.349   | 0.781    |
> | **0.08**                   | 0.941     | 0.357  | 1.329   | 0.796    |
> | **0.1**                    | 0.923     | 0.361  | 1.326   | 0.798    |
>
> **2- Qualitative comparison of SPALE guidance**
>
> Regarding this comment, we would like to refer the reviewer to kindly view the randomly selected samples from different diffusion models in Figure 11 (Qualitative results with different CFG scales), Figure 14 (PixArt-$\Sigma$ results), and Figure 15 (SDXL results) in the appendix.
>
> To further address the reviewer’s comment, we also provide quantitative comparisons of these state-of-the-art models, as shown in the following table, which shows an increase in diversity scores (Con-Vendi, Vendi, In-batch Sim, scores) with only a minor decrease in the evaluated CLIPScore. We also highlight that the kernel distance (KD) evaluation score (evaluating both quality and diversity) and the Authenticity score (with MS-COCO being the reference dataset) have indeed improved in the guided sample generation, indicating that the scores evaluating both quality and diversity have increased.
>
> | Method                  | CLIPScore $\uparrow$ | KD $\times 10^2 \downarrow$ | AuthPct $\uparrow$ | Cond-Vendi Score $\uparrow$  | Vendi Score $\uparrow$ | In-batch Sim.$\times 10^2 \downarrow$ |
> |-------------------------|----------------------|------------------|---------|------------------|-------------|----------------------------|
> | **SDXL**                    | 31.17                | 66.37            | 76.28   |        27.54     |   309.54    | 81.67                      |
> | **SDXL + SPALE**            | 30.47                | 62.62            | 80.92   |          31.17   |   313.80    | 77.75                      |
> | **PixArt-$\Sigma$**         | 31.01                | 65.37            | 78.13   |       26.45      |   307.36    | 83.84                      |
> | **PixArt-$\Sigma$ + SPALE** | 30.66                | 63.86            | 81.48   |       32.14      |   322.25    | 78.81                      |
>
>
> **3- Qualitative comparison of Latent (ours) vs Ambient-based diversity guidance methods**
>
> We would like to highlight that one key property of our proposal is that we apply guidance on the latent space in contrast to the previous works that guide diversity scores in the ambient space (after passing through the CLIP or DINO embeddings). We observed that in large-scale diffusion models, the latent diversity guidance prevents generation artifacts and will generate semantically diverse concepts. In Figure 3 (in the main text), we showed the comparison of Vendi score guidance on the latent space vs. CLIP ambient space on SDXL, which shows that ambient guidance will lead to noticeable noise in the image, rather than true semantic diversity. Additional qualitative results are provided in Figure 12 of the Appendix, which further support this point.

---

> > ### Comment · Reviewer_HGHP · 2025-08-05
> >
> > Thanks to the replies and additional experiments from the authors, most of my concerns have been resolved. However, I still feel that there is room for improvement regarding the performance and general applicability of the proposed method. I will provide my final rating after considering the opinions of the other reviewers.

---

> ### Author Response · Authors · 2025-08-05
>
> We sincerely thank Reviewer HGHP for the feedback on our response and are pleased that our rebuttal was able to address most of the reviewer’s concerns. We will revise the manuscript accordingly to reflect the clarifications provided in our responses.
>
> Regarding the reviewer’s point on the potential for further improvement in the results, we would be grateful if the reviewer could point us to any specific baseline on diversity guidance in diffusion models that they may have in mind. We would like to highlight that our method demonstrates improvements over existing diversity guidance baselines, including CADS [1], Particle Guide [2], and c-VSG [3], as highlighted in quantitative comparisons in Table 3 (Appendix) and supported by the runtime/GPU usages referenced in our response to Reviewer W1na. If Reviewer HGHP has a particular baseline in mind, we would be happy to provide additional experimental results—time permitting—to address it.
>
> ---
>
> [1] Sadat et al. “CADS: Unleashing the Diversity of Diffusion Models through Condition-Annealed Sampling”, ICLR 2024
>
> [2] Corso et al. “Particle Guidance: non-I.I.D. Diverse Sampling with Diffusion Models”, ICLR 2024
>
> [3] Askari Hemmat et al. “Improving Geo-diversity of Generated Images with Contextualized Vendi Score Guidance”, ECCV 2024

---

### Official Review · Reviewer_wxtJ · 2025-07-07

**Clarity:** 3
**Significance:** 3
**Originality:** 3
**Rating:** 4
**Confidence:** 4

**Summary:**

This paper considered the problem of ensuring prompt-aware diversity when guiding diffusion models. The main idea is based on using the matrix-based Rényi entropy (RKE score), which measures the kernel-based similarity between a class of $n$ prompts and associated images. By using a gradient-based optimization approach, the authors designed an algorithm that only needs to compute the gradient of the conditional RKE score, which further reduced the computational cost from $O(n^3)$ to $O(n)$. Numerical experiments on various text-to-image diffusion models are provided to demonstrate the proposed method's effectiveness in terms of prompting the diversity of generated images.

**Questions:**

The reviewer is satisfied with the quality of the manuscript and has just one follow-up question. Specifically, the authors of [10] have made a link between their proposed guidance method with other kernel-based/ensemble-based sampling methods like the Stein Variational Gradient Descent (SVGD) [11]. Would it be possible for the authors to briefly comment on how the method proposed in this manuscript may relate to other ensemble-based/kernel-based sampling methods in literature? (maybe like that of [12,13] - Is there a gradient-flow based interpretation of the proposed method?) Alternatively, would it be possible for the authors to provide some insights on how one may perform theoretical analysis (like that of [14,15,16]) on the proposed guidance methods, maybe at least for the case of Gaussian mixtures?

**Ethical Concerns:**

["NO or VERY MINOR ethics concerns only"]

**Final Justification:**

The reviewer has checked all of the reviews and the authors' responses. Overall, most issues related to both theoretical and practical aspects of the proposed method have been resolved. Therefore, given that the additional experimental results attached in the authors' rebuttal haven't been added to the main text yet, the reviewer is recommending "Borderline Accept" by maintaining a score of 4.

**Limitations:**

The reviewer is satisfied with the limitations that the authors briefly discussed at the end of the manuscript. Specifically, the authors mentioned that currently the methodology can only be used for image-based diffusion models. Hence, one of the future directions is to investigate whether the methodology can be generalized to other modalities like video and text.

References:

[1] Singhal, R., Horvitz, Z., Teehan, R., Ren, M., Yu, Z., McKeown, K., & Ranganath, R. (2025). A general framework for inference-time scaling and steering of diffusion models. arXiv preprint arXiv:2501.06848.

[2] Chen, H., Ren, Y., Min, M. R., Ying, L., & Izzo, Z. (2025). Solving inverse problems via diffusion-based priors: An approximation-free ensemble sampling approach. arXiv preprint arXiv:2506.03979.

[3] Skreta, M., Akhound-Sadegh, T., Ohanesian, V., Bondesan, R., Aspuru-Guzik, A., Doucet, A., ... & Neklyudov, K. (2025). Feynman-kac correctors in diffusion: Annealing, guidance, and product of experts. arXiv preprint arXiv:2503.02819.

[4] Wu, L., Trippe, B., Naesseth, C., Blei, D., & Cunningham, J. P. (2023). Practical and asymptotically exact conditional sampling in diffusion models. Advances in Neural Information Processing Systems (NeurIPS), 36, 31372-31403.

[5] Trippe, B. L., Yim, J., Tischer, D., Baker, D., Broderick, T., Barzilay, R., & Jaakkola, T. (2022). Diffusion probabilistic modeling of protein backbones in 3d for the motif-scaffolding problem. In The Eleventh International Conference on Learning Representations (ICLR), 2023.

[6] Cardoso, G., Idrissi, Y. J. E., Corff, S. L., & Moulines, E. (2023). Monte Carlo guided diffusion for Bayesian linear inverse problems. In The Twelfth International Conference on Learning Representations (ICLR), 2024.

[7] Dou, Z., & Song, Y. (2024). Diffusion posterior sampling for linear inverse problem solving: A filtering perspective. In The Twelfth International Conference on Learning Representations (ICLR), 2024.

[8] Kim, S., M. Kim, and D. Park (2025). Test-time alignment of diffusion models without reward over-optimization. In The Thirteenth International Conference on Learning Representations (ICLR), 2025.

[9] Guo, Y., Yuan, H., Yang, Y., Chen, M., & Wang, M. (2024). Gradient guidance for diffusion models: An optimization perspective. arXiv preprint arXiv:2404.14743.

[10] Corso, G., Xu, Y., De Bortoli, V., Barzilay, R., & Jaakkola, T. (2023). Particle guidance: non-iid diverse sampling with diffusion models. In The Twelfth International Conference on Learning Representations (ICLR), 2024.

[11] Liu, Q., & Wang, D. (2016). Stein variational gradient descent: A general purpose bayesian inference algorithm. Advances in Neural Information Processing Systems (NeurIPS), 29.

[12] Wild, V. D., Ghalebikesabi, S., Sejdinovic, D., & Knoblauch, J. (2023). A rigorous link between deep ensembles and (variational) Bayesian methods. Advances in Neural Information Processing Systems (NeurIPS), 36, 39782-39811.

[13] Li, L., Liu, Q., Korba, A., Yurochkin, M., & Solomon, J. (2022). Sampling with mollified interaction energy descent. In The Eleventh International Conference on Learning Representations (ICLR), 2023.

[14] Chidambaram, M., Gatmiry, K., Chen, S., Lee, H., & Lu, J. (2024). What does guidance do? a fine-grained analysis in a simple setting. Advances in Neural Information Processing Systems (NeurIPS), 37.

[15] Wu, Y., Chen, M., Li, Z., Wang, M., & Wei, Y. (2024). Theoretical insights for diffusion guidance: A case study for gaussian mixture models. International Conference on Machine Learning (ICML), PMLR, 2024.

[16] Li, G., & Jiao, Y. (2025). Provable Efficiency of Guidance in Diffusion Models for General Data Distribution. arXiv preprint arXiv:2505.01382.

**Quality:**

3

**Strengths And Weaknesses:**

Pros: The reviewer finds that the paper is presented in a clear way for the readers to comprehend. For all theoretical arguments, the authors have included detailed proof in the appendix. Also, several large-scale models like Stable Diffusion are used to test the validity of the proposed method. Both experimental and implementational details are provided.

Cons: The reviewer found no significant weaknesses of the paper. However, the authors are encouraged to perform a more comprehensive literature review for related work on guidance. For instance, to the best of the reviewer's knowledge, a popular class of work have developed guidance methods based on sequential monte carlo (SMC), such as [1,2,3,4,5,6,7,8] (Though some work might have studied different problems like inference-time scaling or Bayesian inverse problems, the formulation is similar). Other work that the authors might have missed citing include [9], which discusses an optimization based formulation of guidance methods.

---

> ### Author Rebuttal · Authors · 2025-07-31
>
> We would like to thank Reviewer wxtJ for the constructive and thoughtful feedback and are delighted to hear that the reviewer found our work “clearly presented” and “well-supported by theoretical proofs and large-scale experiments”. Below is our response and clarifications regarding the questions and comments in the review:
>
> **1- Literature review for related works on guidance**
>
> We thank Reviewer wxtJ for mentioning the related works on guidance in diffusion models. As pointed out in the review, a more comprehensive discussion of guidance methods, particularly those based on Sequential Monte Carlo (SMC) and optimization formulations, will definitely position our work better in this literature. We will revise the related work section to include and discuss the mentioned references.
>
> **2- Relation to other kernel-based sampling methods**
>
> We thank the reviewer for the valuable feedback and for raising the connection to other kernel-based sampling methods. We appreciate this opportunity to clarify the theoretical distinctions and contributions of SPALE.
>
> Our answer to the reviewer’s question is positive, and there is a gradient-flow-based interpretation of our method. SPALE, along with Stein Variational Gradient Descent (SVGD)[1], Particle Guidance (PG)[2], and the ensemble-based/kernel-based sampling methods in [3, 4], can be understood as an interacting particle system that follows a gradient flow to minimize a specific energy function:
> $$\mathrm{d}\mathbf{z}^{(n)} = \left[ - \mathbf{f}(\mathbf{z}^{(n)}, t') + \frac{g^2(t')}{2} \left( \nabla_{\mathbf{z}^{(n)}} \log p_{t'}(\mathbf{z}^{(n)}) - \eta \nabla_{\mathbf{z}^{(n)}} L_{\text{Cond-IRKE}}(\mathbf{z}_1, \dots, \mathbf{z}_n) \right) \right] \mathrm{d}t + g(t') \mathrm{d}\mathbf{w}$$
> While previous methods define a gradient flow to optimize for sample quality and **prompt-unaware** diversity, our proposed SPALE defines a flow to minimize a Conditional Inverse-Rényi Kernel Entropy (Cond-IRKE). The core idea of SPALE lies in the novelty of the energy function it minimizes, thus represents a forward by introducing *prompt-awareness* and improving *scalability*.
>
> **Prompt-Awareness**. Previous methods like Particle Guidance (PG)[10] employ a prompt-unaware repulsion mechanism. Their guidance potential $\log \Phi_{t'}(\mathbf{z}_1, \dots, \mathbf{z}_n)$ treats all samples in a batch of different prompts equally, applying a uniform repulsion force between every pair regardless of their corresponding prompts. This uniform repulsion limits the computational efficiency, and it is not completely aligned with the goal of generating diverse outputs within related contexts, i.e., prompt categories. SPALE introduces a prompt-aware guidance mechanism, which is the key difference from previous methods. By optimizing the **Conditional Rényi Kernel Entropy** during sampling, our guidance potential is defined by:
> $$
> \nabla\_{z^{(n)}} L\_{\text{Cond-IRKE}} \propto \mathbb{E}\_{ {(\mathbf{z}', \mathbf{y}') \sim P\_{n-1}} } \left[ k\_Z(\mathbf{z}', \mathbf{z}^{(n)}) \cdot k\_Y(\mathbf{y}', \mathbf{y}^{(n)}) \nabla\_{z^{(n)}} k\_Z(\mathbf{z}', \mathbf{z}^{(n)}) \right]
> $$
>
> **Linear-Time Computational Scalability**. We would like to clarify that in studying the scalability of the proposed SPALE method, we specifically compare it with other diversity-score-based guidance methods, including the Vendi score guidance. The computation of the Vendi score and its gradient requires $O(n^3)$ computations for $n$ generated samples, which enforces considering short windows for applying the diversity guidance in generating a moderately large number of data. On the other hand, the proposed SPALE method switches to the RKE score whose gradient can be calculated with $O(n)$ computations, which allows us to scale the diversity score-guided sample generation to several thousands. We would like to highlight that as shown Theorem 1 in [5], the RKE (entropy order-2) is the only entropy score with such a property in the family of Renyi entropy measures. We will make this point on the scalability of SPALE in comparison to the Vendi score diversity guidance clear.
>
> **3- Insights of theoretical analysis of RKE Guidance**
>
> As previously discussed, our guidance algorithm follows the gradient-flow-based sampling formulations, and we provide a theoretical insight into interpreting how Rényi Kernel Entropy (RKE) score[6] and Conditional RKE score[7] enhance the sample diversity.
>
> The main insight is that the order-2 RKE score serves as a differentiable objective for counting the modes of a particle distribution, and [6] provides a theoretical analysis about the RKE score of the kernel covariance matrix of mixtures of Gaussian and sub-Gaussian components. Furthermore, the Conditional RKE score in [7] quantifies the internal diversity conditioning on the prompt categories.
>
> SPALE defines a gradient flow that optimizes the Conditional RKE score. This process sequentially drives samples to cover distinct modes, while the conditional mechanism ensures the prompt-aware diversity guidance potential. The dynamic nature of this sequential generation scheme makes it analytically challenging to characterize the final joint distribution of the samples. In the revised version, we will include the above discussion on how SPALE achieves prompt-aware diversity via RKE score and Conditional RKE score for future analysis.
>
> ---
>
> [1] Liu, Q., & Wang, D. (2016). Stein variational gradient descent: A general purpose bayesian inference algorithm. Advances in neural information processing systems, 29.
>
> [2] Corso, G., Xu, Y., De Bortoli, V., Barzilay, R., & Jaakkola, T. S. Particle Guidance: non-IID Diverse Sampling with Diffusion Models. In The Twelfth International Conference on Learning Representations.
>
> [3] Wild, V. D., Ghalebikesabi, S., Sejdinovic, D., & Knoblauch, J. (2023). A rigorous link between deep ensembles and (variational) bayesian methods. Advances in Neural Information Processing Systems, 36, 39782-39811.
>
> [4] Li, L., Korba, A., Yurochkin, M., & Solomon, J. Sampling with Mollified Interaction Energy Descent. In The Eleventh International Conference on Learning Representations.
>
> [5] Ospanov, A., & Farnia, F. Do Vendi Scores Converge with Finite Samples? Truncated Vendi Score for Finite-Sample Convergence Guarantees. In The 41st Conference on Uncertainty in Artificial Intelligence.
>
> [6] Jalali, M., Li, C. T., & Farnia, F. (2023). An information-theoretic evaluation of generative models in learning multi-modal distributions. Advances in Neural Information Processing Systems, 36, 9931-9943.
>
> [7] Jalali, M., Ospanov, A., Gohari, A., & Farnia, F. (2024). Conditional Vendi Score: An information-theoretic approach to diversity evaluation of prompt-based generative models. arXiv preprint arXiv:2411.02817.

---

> > ### Comment · Reviewer_wxtJ · 2025-08-05
> > **Response to authors**
> >
> > The reviewer would like to thank the authors for their detailed response, most of which the reviewer is satisfied with. Therefore, the reviewer will be keeping the score and recommending acceptance. Moreover, the authors are highly encouraged to take suggestions raised in most of the reviews into consideration while finalizing the camera-ready version of the manuscript [Examples include but not limit to a more comprehensive literature review (guidance methods based on SMC), interpretation of the proposed methods via gradient flow, or future work on the theoretical analysis of the proposed method.]

---

> > > ### Author Response · Authors · 2025-08-05
> > >
> > > We sincerely thank Reviewer wxtJ for the feedback on our response and are pleased that the reviewer is satisfied with most of our rebuttal. As we mentioned in our response, we will revise the manuscript accordingly to reflect the points and clarifications provided in our responses to this and other reviews.

---

### Note · Authors · 2025-08-13

We sincerely thank the reviewers for their insightful feedback on our work and rebuttal. Below, we summarize the key points we aimed to highlight during the discussion phase:

**Context and motivation**. Diversity guidance in diffusion sample generation has been actively studied in recent works, including CADS [1], Particle Guide [2], and c-VSG [3]. While these methods have shown promising results for *standard* diffusion sampling, two critical challenges remain:

1. **Prompt-aware diversity guidance** in evaluating diversity across semantically similar input prompts.
2. **Scalability** in ensuring computational feasibility when generating hundreds or thousands of samples.

Our proposed RKE-based diversity guidance is designed to address these challenges.

**Prompt-aware extension via Conditional-RKE**. The proposed RKE-score diversity measure naturally extends to prompt-aware diversity guidance through the *Conditional-RKE* score, which evaluates the prompt-conditioned entropy of generated data. This is a key component of our framework, leading to improved diversity within sets of semantically similar prompts.

**Scalability of SPALE guidance**. SPALE achieves scalability through two techniques:

1. Using the **Inverse-RKE** diversity score whose gradient decomposes into gradient sums of kernel distances involving only $n$ terms at iteration $n$.

2. Implementing I-RKE diversity guidance in the **latent space** of a *Latent Diffusion Model (LDM)*, significantly reducing time and memory complexity compared to baselines. In our response to Reviewer W1na, we included a detailed scalability comparison.

**Diversity–fidelity performance scores**. Compared to CADS, Particle Guide, and c-VSG (baselines), our proposed SPALE method could achieve higher diversity scores and better scalability, *without compromising quality metrics* including kernel distance (KD), authenticity, and CLIP-Score. In fact, fidelity scores even improve over the mentioned diversity guidance methods, as applying guidance in the latent space reduces visual artifacts. A full comparison of diversity and fidelity results is provided in Table 3 of the Appendix.


[1] Sadat et al. “CADS: Unleashing the Diversity of Diffusion Models through Condition-Annealed Sampling”, ICLR 2024

[2] Corso et al. “Particle Guidance: non-I.I.D. Diverse Sampling with Diffusion Models”, ICLR 2024

[3] Hemmat et al. “Improving Geo-diversity of Generated Images with Contextualized Vendi Score Guidance”, ECCV 2024

---

### Decision · Program_Chairs · 2025-09-17

**Decision:**

Accept (poster)

**Comment:**

This paper introduces Scalable Prompt-Aware Latent Entropy-based Diversity Guidance for prompt-aware diversity guidance. Reviewers are satisfied with the writing of the paper, the theoretical rigor, and the performance of the proposed method. Most of the raised concerns are addressed during the rebuttal. In the end, all the reviewers recommend accept. Therefore, I'm recommending an accepetance and encourage the authors to keep improving the current submission based on the feedbacks from reviewers.